# SAEs-BrainMap: Unveiling the Emergence of Specialized Concepts in Deep Models via Brain Alignment

Ziming Mao [1 2]   Jia Xu [1]   Wenxuan Pan [2]   Mufan Xue [3]   Yaochu Jin [2]   Guoyuan Yang [3 4]

## Abstract

Understanding the internal mechanisms of deep neural networks remains a significant challenge, particularly in elucidating how generic visual concepts emerge within latent spaces. In this work, we propose SAEs-BrainMap, a novel framework that utilizes human brain activation patterns from the ventral visual pathway as objective probes to guide the identification of features decomposed by Sparse Autoencoders (SAEs). Our quantitative and qualitative empirical results demonstrate a robust representational alignment between sparse model features and biological Regions of Interests (ROIs), confirming the feasibility of utilizing brain signals to characterize model functionality. By leveraging this alignment, we trace the hierarchical trajectory of generic concepts cross layers and utilize the brain's hierarchical structure to visualize the model's global processing flow, providing novel insights into model interpretability. Our code is available at this repository.

## 1. Introduction

Deep Neural Networks (DNNs), particularly Transformer-based architectures, have achieved remarkable success across various domains in computer vision. However, due to their inherent "black-box" nature, deploying these models in safety-critical and transparency-sensitive domains remains challenging (Tripicchio & D'Avella, 2020). To bridge this gap, the field of Explainable Artificial Intelligence (XAI) has experienced rapid growth in recent years, with concept-based methods (Kim et al., 2018) emerging as a powerful framework to decode models' cognitive mechanisms into human-understandable concepts.

Existing concept-based approaches can be categorized into neuron-based and representation-based methods (Zou et al., 2023). The former, such as Network Dissection (Bau et al., 2017), Describe-and-Dissect (DnD) (Bai et al., 2024), and Granular Concept Circuits (GCC) (Kwon et al., 2025), interpret model functions in terms of neurons and circuits, treating individual neurons as the fundamental unit(Oikarinen & Weng, 2022). The latter places vector representations at the center of analysis, positing that visual concepts are encoded as directions in the model's latent space, as in Visual Concept Connectome (VCC) (Kowal et al., 2024) and Sparse Autoencoders (SAEs) (Huben et al., 2024).

While these methods have deepened our understanding of how a certain category is learned in the models, they still face significant challenges in elucidating the emergence procedure of generic concepts. For neuron-based methods, the primary obstacle is the neurons' polysemantic nature, which complicates the localization of fine-grained concept emergence due to feature entanglement (Huben et al., 2024; Zou et al., 2023; Thasarathan et al., 2025). Conversely, for representation-based methods, the challenge lies in semantic alignment. Identifying the specific vector direction corresponding to a generic concept is hard due to the opacity of the latent space (Kowal et al., 2024) and the difficulty of training suitable probes, such as classification heads (Fel et al., 2025), to track concepts consistently across layers.

To address these limitations and elucidate how deep visual models construct generic concepts, such as 'Faces' and 'Places', we draw motivation from recent neuroscience studies like BrainSCUBA (Luo et al., 2024), BrainACTIV (Cerdas et al., 2024) and BrainExplore (Wasserman et al., 2025). While these works demonstrate that model latent vectors can interpret brain functions, we seek to explore the feasibility of inverting this paradigm. Specifically, we investigate whether activations from brain ROIs—given their capacity to process broad and abstract information (He et al., 2022)—can serve as objective probes to characterize the emergence of generic visual concepts within the models.

---

[1] School of Computer Science and Technology, Beijing Institute of Technology, Beijing, China [2] School of engineering, Westlake University, Hangzhou, China [3] School of Interdisciplinary Science, Beijing Institute of Technology, Beijing, China [4] School of Medical Science and Engineering, Beijing Institute of Technology, Beijing, China. Correspondence to: Guoyuan Yang <yanggy@bit.edu.cn>, Yaochu Jin <jinyaochu@westlake.edu.cn>.

*Proceedings of the $43^{rd}$ International Conference on Machine Learning*, Seoul, South Korea. PMLR 306, 2026. Copyright 2026 by the author(s).

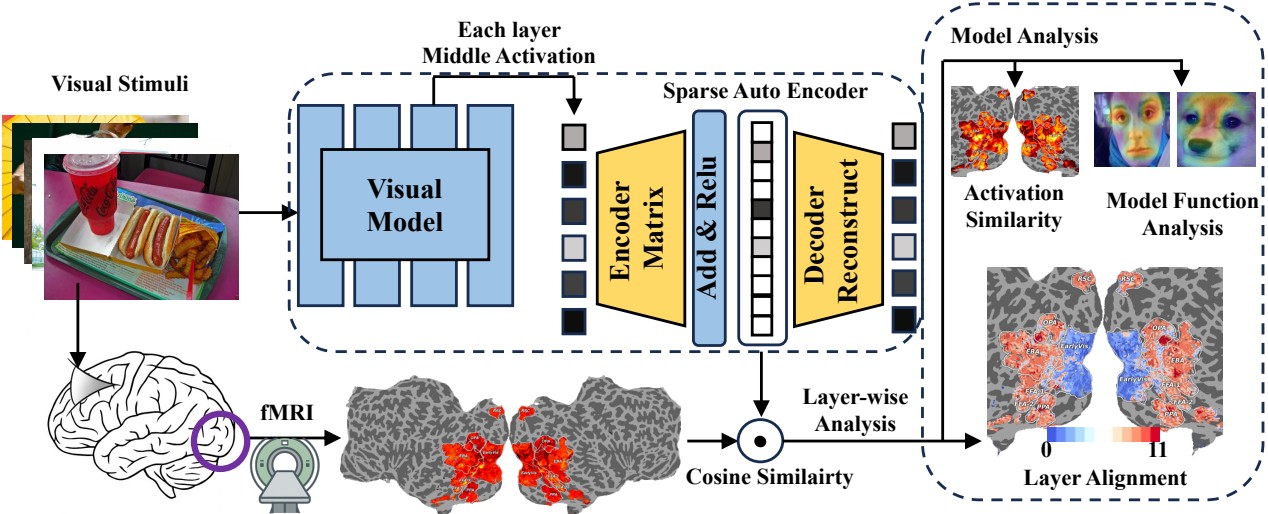

*Figure 1.* Structure of SAEs-BrainMap. We first train layer-wise SAEs to capture the monosemantic features existing in each layer, then we utilize the brain activation as a objective probe, to investigate the emergence of generic concepts in the model's latent space. In the end, we mapping the model layers onto the brain cortex to find out the most functional correlated layer for each region.

In this work, we utilize brain fMRI signals to guide the selection of specific features decomposed by SAEs in a data-driven manner, exploring the emergence mechanism of generic concepts that correlate with target brain ROIs (Figure 1). The main contributions of this paper are summarized as follows. (1) We propose SAEs-BrainMap, which pioneers the utilization of brain activation to investigate layer-wise SAEs functions. (2) We demonstrate the representational alignment between model SAEs features and brain voxels—both activationally (max 0.72 cosine similarity) and structurally (max 0.69 RSA score)—thereby validating the feasibility of employing the brain as a functional probe. (3) Leveraging SAEs-BrainMap, we unveil the emergence trajectory of generic concepts via brain-model alignment.

## 2. Related Work

**Concept-based Interpretability** decompose model internal representations through the human-interpretable units rather than raw input features (Kim et al., 2018). It can be separated by the representation of the concept into neuron-based and representation-based methods. Neuron-based method including CRP (Achtibat et al., 2023), FALCON (Kalibhat et al., 2023), RDR (Chang et al., 2024), Dissection series like Network Dissect (Bau et al., 2017), CLIP-Dissect (Oikarinen & Weng, 2022) and DnD (Bai et al., 2024). They consider the model neurons as the unit of recognition and figure out several labels to describing the main function of this units. Moreover, methods like ADVC (Rajaram et al., 2024) and GCC (Kwon et al., 2025) finding the circuits composed of neurons for a certain function. Representation-based consider the concept as vectors, VCC (Kowal et al., 2024) utilize cluster method to find the representation vectors in model latent space, while SAEs decompose model activations into linear combinations of sparse, monosemantic sparse features (Gorton, 2024; Huben et al., 2024; Thasarathan et al., 2025; Fel et al., 2025). In this work, we will utilize the SAEs to extract the concept specific representation vectors in model's latent spaces.

**Ventral Visual Pathway** is a hierarchical processing stream responsible for the formation of visual concepts (DiCarlo & Cox, 2007; Goodale et al., 1994; He et al., 2022; Murata et al., 2000). Visual information starts from the early visual cortex V1, and progresses through V2, V3, and hV4. Voxels along this pathway exhibit increasingly larger receptive fields (Dumoulin & Wandell, 2008) and transit from encoding low-level visual features such as edges and curvature (Awang, 1962; Carandini et al., 2005) to processing more complex attributes, including shape, color, texture, and depth (Coggan et al., 2017; Levitt et al., 1994; Freeman et al., 2013). Subsequently, high-level semantic understanding is accomplished in the high-level visual cortex. Notable examples include the fusiform face area (FFA) for face selectivity (Kanwisher et al., 1997), the retrosplenial cortex (RSC) for scene selectivity (Epstein & Kanwisher, 1998), the extrastriate body area (EBA) for body selectivity (Downing et al., 2001), the visual word form area (VWFA) for word selectivity (Cohen et al., 2000), and a region showing selectivity for food-related stimuli (Khosla et al., 2022). Our study mainly focus on these ROIs, as illustrated in Figure 2.

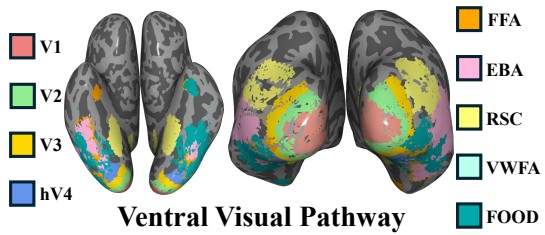

*Figure 2.* Visualization of S5 ventral visual pathway ROIs.

*Table 1.* Quantitative brain-model alignment results. We report Max/Mean Cosine Similarity and RSA scores measuring the correspondence between fMRI data and model features (Neurons, CLS-trained SAEs, and Patch-trained SAEs) for S5 across four different models. The CLIP in the table refers to the CLIP ViT-B/16.

| MODEL | MAX SIMI | MEAN SIMI | RSA SCORE |
|---|---|---|---|
| **CLS SAEs** | | | |
| CLIP | 0.7247 | 0.2811 | 0.580 |
| DINOV2 | 0.7172 | 0.2705 | 0.686 |
| MAE | 0.6198 | 0.2582 | 0.646 |
| IMAGENET | 0.6431 | 0.2495 | 0.630 |
| **PATCH SAEs** | | | |
| CLIP | 0.6858 | 0.2708 | 0.596 |
| DINOV2 | 0.6517 | 0.2670 | 0.592 |
| MAE | 0.6376 | 0.2327 | 0.586 |
| IMAGENET | 0.6019 | 0.2392 | 0.656 |
| **RAW NEURONS** | | | |
| CLIP | 0.5875 | 0.2354 | 0.511 |
| DINOV2 | 0.6961 | 0.2421 | 0.579 |
| MAE | 0.5108 | 0.2087 | 0.551 |
| IMAGENET | 0.5319 | 0.2245 | 0.596 |

**Brain Function Interpret Method** primarily aim to elucidate the functional roles of individual voxels (van Dyck et al., 2025). These approaches largely rely on Brain Encoders, which predict voxel-level activations from the model's image embeddings (Wang et al., 2023). Building on this, recent works (Luo et al., 2024; Cerdas et al., 2024) utilize brain encoders to identify the optimal vector representation for each voxel within the model's latent space, subsequently leveraging diffusion models to evaluate the semantic fidelity of these representations. We draw inspiration from these methodologies and brain-model alignment works (Liu et al., 2023; Zhao et al., 2023), which substantiate the robust similarity in both activation patterns and representational structures between the two systems.

## 3. Method

In this section, we first present several preliminaries for sparse autoencoders and vision transformers. Subsequently,

we introduce the pipeline for extracting biologically aligned features, detailing our selection and evaluation metrics.

### 3.1. Preliminary

**Sparse Autoencoders (SAEs)** (Huben et al., 2024) aim to reconstruct model activations using an encoder-decoder architecture. Let $\mathcal{I}$ be a dataset consisting of $n$ images, and let $f : \mathcal{I} \to A$ denote a ViT-based model embedding process. The image embeddings are $A \in \mathbb{R}^{n \times (L+1) \times d}$, where $d$ is the embedding dimension, $L + 1$ refers to the total sequence length, comprising $L$ patch tokens and an [CLS] token. The SAEs learns a linear encoder $\Psi_\theta(\cdot)$ that maps the dense embeddings $A$ to a set of $k$ sparse feature activations $Z \in \mathbb{R}^{n \times (L+1) \times k}$. A linear decoder $\Phi_\theta(\cdot)$, which shares the same linear weight $W^T \in \mathbb{R}^{k \times d}$ with the encoder, reconstructs the original embeddings $\hat{A}$ from the sparse activation $Z$. This process is formulated in Equations 1, 2:

$$Z = \Psi_\theta(A) = ReLU(WA + b) \tag{1}$$

$$\hat{A} = \Phi_\theta(Z) = W^T Z = \sum_{i=0}^{k-1} W_i^T Z_i, \tag{2}$$

where $W \in \mathbb{R}^{d \times k}$ and $b \in \mathbb{R}^k$ are the learned parameters. Each unit corresponding to a column in $W$ serves as a sparse feature vector, collectively forming the SAE feature dictionary (Fel et al., 2025). The reconstructed embedding $\hat{A}$ is thus a linear combination of these dictionary units weighted by their positive activations. The training process aims to minimize the loss function $\mathcal{L}$ in Equation 3, where the hyperparameter $\alpha$ controls the sparsity of the reconstruction.

$$\mathcal{L}(A, \hat{A}, Z) = \|\hat{A} - A\|_2^2 + \alpha \|Z\|_1 \tag{3}$$

We train SAEs for each layer of a deep learning model. For the $j$-th layer, let $A^{(j)}$ denote the layer output, $\Psi_\theta^{(j)}$ the encoder, and $Z^{(j)}$ the corresponding SAE activations.

**Representational Similarity Analysis (RSA)** (Kriegeskorte et al., 2008) provides a framework for characterizing representational geometries and quantifying the similarity between distinct systems. For a set of $n$ stimuli, we first compute a Representational Similarity Matrix (RSM) $M \in \mathbb{R}^{n \times n}$ for each system as (Conwell et al., 2024), where each entry $M_{ij}$ measures the correlation between the system's response to stimuli $i$ and $j$. RSA then quantifies the similarity by comparing the two RSMs. In this study, we first select a SAEs' feature for each voxel that exhibit the highest similarity to the voxel's fMRI signal. Then, we apply RSA to compare the representational geometries between the brain fMRI signals $Q \in \mathbb{R}^{n \times v}$ and the average activations of voxel-wise selected SAEs' features $Z' \in \mathbb{R}^{n \times v}$, where $v$ is the number of voxels and $Z'$ denotes the features' average activations across all patches. The RSM is computed using **Pearson correlation**, the RSA score is

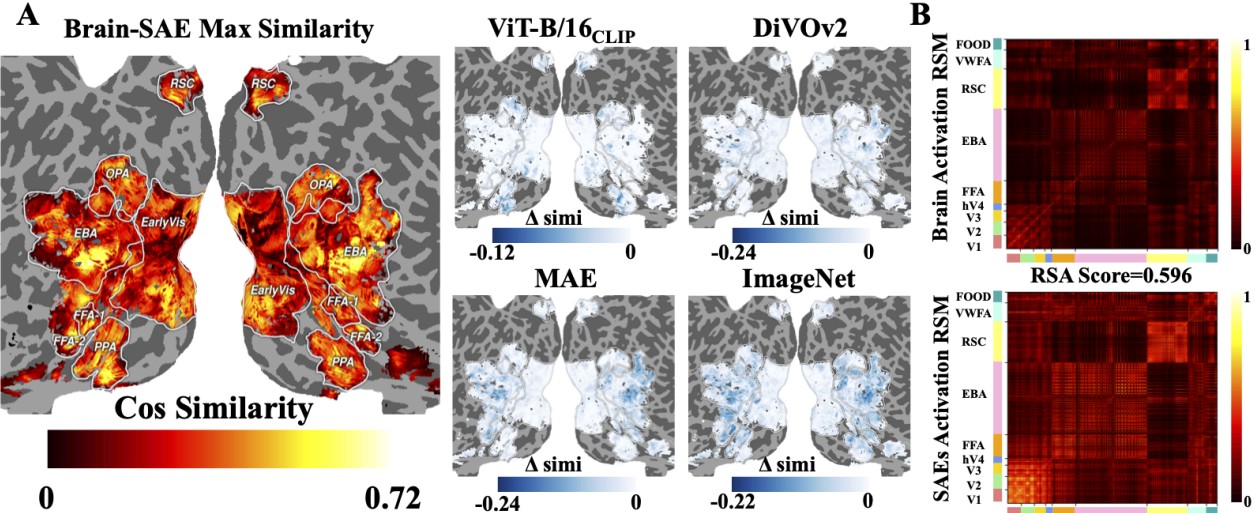

*Figure 3.* This figure illustrates the activation correlation between SAEs and fMRI signals for S5. **A** Brain-SAE activation similarity. The left cortex map denote the maximum cosine similarity per voxel found across all models and layers, with additional plots showing the deviation of individual models relative to this peak, indicating the widespread alignment across models. **B** RSA analysis on ViT-B/16$_{CLIP}$ for S5. Prior to computing the RSA, the best-matching SAEs feature was selected for each voxel. The selected SAEs activation RSM (bottom) is compared with the Brain activation RSM (top) using Spearman correlation, yielding a RSA score of 0.596.

computed using **Spearman correlation** and is defined as:

$$\text{RSA Score} = r_{spearman}(sim(Q, Q), sim(Z^{'}, Z^{'})) \quad (4)$$

### 3.2. Brain-SAEs Correlation and Evaluation

**ROI-based SAEs' Features Selection**. We aim to leverage neural representations to elucidate how the model encodes specific semantic concepts. Given the challenges in interpreting fine-grained voxel-wise functions, we conduct our experiments at the ROI level, focusing on five high-level visual areas with well-established selectivities: FFA (faces), EBA (bodies), RSC (scenes), VWFA (words), and regions selective for food. Then we compute the similarity matrix $M_{sim} = \text{cosine}(Q_{ROI}, Z^{(j)})$, representing the alignment between a certain ROI fMRI signals $Q_{ROI} \in \mathbb{R}^{n \times v_{ROI}}$ and the $j$-th layer SAEs features. Subsequently, we identify the top-100 features that exhibit the **highest average similarity across the entire ROI**. These units are denoted as the candidate that functionally align with the target ROI in this layer. The procedure is applied to all layers of the model, aiming to trace the emergence of certain concepts. Note that the experiment is subject-specific and is performed on their individual visual stimuli dataset, as introduced in Section 4.1.

**Candidate Feature Evaluation**. Following the selection of candidate concept-specific features across all layers, we first retrieve the set of top-$k$ images, denoted as $\mathcal{I}_c^i = \{x_1, \ldots, x_k\}$, that maximally activate the candidate

feature $i$ (selected by ROI $c$) from the ImageNet-1K test set (Russakovsky et al., 2015). Subsequently, we employ a CLIP-based classification protocol adapted from (Yu et al., 2026) to evaluate both the selectivity stability and semantic alignment of these features. We utilize a concept set $\mathcal{C}_{total}$ containing the five target ROI functional concepts and a uncorrelated control concepts. For each concept $c^{'} \in \mathcal{C}_{total}$, we construct a corresponding text prompt set $T_{c^{'}}$ (see Appendix D). For each image $x_n \in \mathcal{I}_c^i$, we obtain the predicted concept $\hat{y}_n$ by maximizing the similarity between the CLIP image embedding and the text prompt embeddings. Let $c^*$ denote the functional concept associated with the target ROI $c$. The stability score $S_i$ for feature $i$ is then calculated as: $S_i = \frac{1}{k} \sum_{n=1}^{k} \mathbb{I}(\hat{y}_n = c^*)$, where $\mathbb{I}(\cdot)$ is the indicator function. We defined a selected feature as stable and aligned with the ROI's function if $S_i > \beta$, where $\beta$ is a threshold.

## 4. Experiment

In this section, we first investigate the alignment between fMRI signals and SAEs activations, revealing a strong correspondence in their patterns. We then demonstrate that, based on ROI-level activation patterns, we can identify a subset of SAEs features that exhibit functional behaviors consistent with biologically informed priors. Depending on these ROI-selected features, we further analyze how deep models progressively develop representations for specific visual concepts across layers. Finally, we present a global

visualization of the model's functional organization by mapping its layers onto the brain cortex, providing a multi-level perspective on model's hierarchical visual processing.

## 4.1. Setup

The fMRI dataset we use in the experiment is Nature Scenes Dataset (NSD) (Allen et al., 2022), each of the subject was expected to view 10,000 natural scene images selected from COCO (Lin et al., 2014) for three times, while all the subject share the same set of 1000 images and an independent set of 9000 images. The four visual models in the experiment are CLIP ViT-16/B (Radford et al., 2021), MAE ViT-B/16 (He et al., 2022), DiNOv2 ViT-B/14 (Oquab et al., 2023), and ImageNet trained ViT-16/B (Dosovitskiy et al., 2020) (the model is named ImageNet in the paper) loaded from pytorch (Paszke et al., 2019). We trained separate SAEs for the [CLS] token and the patch tokens on the ImageNet-1k training set (Russakovsky et al., 2015). This procedure was applied to every layer across all models following (Huben et al., 2024), more details can be find in Appendix C.

## 4.2. Brain-Model Similarity

To investigate the feasibility of using brain functional priors to interpret model function, we calculate the activation correlation between fMRI voxel signals and the activations of both model neurons and SAEs first. We perform the alignment using two methods: at a fine-grained level, by directly calculating the cosine similarity between fMRI and model activations; and at a structural level, by employing RSA to measure their representational geometries similarity. The results indicate that models aligned well with the ventral visual pathway at both the activation and structural levels (Figure 3).

**Brain-Model Activation Cosine Similarity**. For each model and layer, we extract activation matrices corresponding to CLS-trained SAEs ($Z_{cls}$), patch-trained SAEs ($Z_{patch}$), and raw model neurons ($N$) in response to independent stimuli for each subject. Taking CLIP ViT-B/16 as an example, the dimensions are defined as $Z_{cls} \in \mathbb{R}^{n \times 1 \times k}$, $Z_{patch} \in \mathbb{R}^{n \times 196 \times k}$, and $N \in \mathbb{R}^{n \times 197 \times 768}$. Prior to analysis, we average $Z_{patch}$ and $N$ along the patch dimension and standardize all activations across the batch dimension. We then measure the alignment between the processed model activations and the brain activation matrix $Q \in \mathbb{R}^{n \times v}$ using cosine similarity. By selecting the peak correlation for each voxel across all layers, we observed that the model aligns significantly with human brain responses, achieving a peak mean correlation of 0.28. It is a strong correlation due to the low signal-to-noise ratio inherent in fMRI data and we make a further discussion in Appendix E. Notably, as shown in Table 1, SAEs consistently outperform raw model neurons in terms of activation similarity across all four mod-

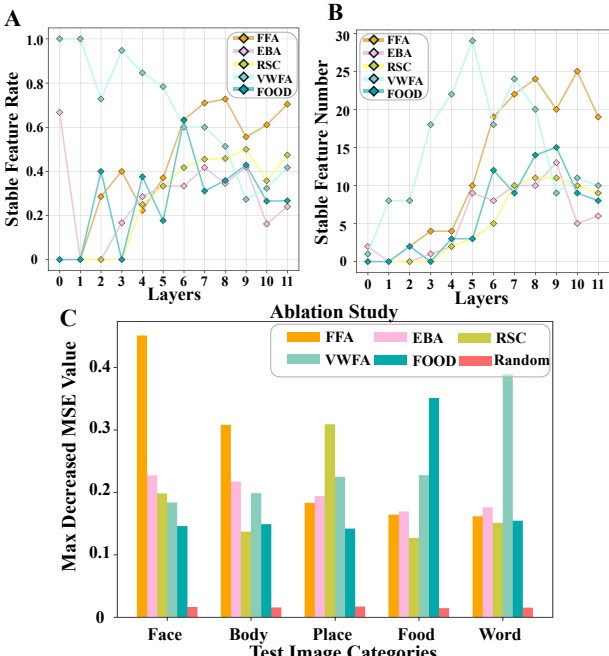

*Figure 4.* ROI-based selected feature evaluation for ViT-B/16$_{CLIP}$, S5. **A** Ratio of functionally aligned features to the total count of stable features, higher ratio indicates higher specificity of our methods. **B** Layer-wise count of stable and functionally aligned features in top-100 features for each ROI, higher number indicates more feature is functionally aligned with the corresponding ROI. **C** Ablation analysis of functionally aligned features. We visualize the maximum normalized MSE loss computed across all model layers for images from five target categories. These further demonstrate these features are functionally aligned with the brain's priors.

els. Figure 3A presents a cortical surface visualization of the voxel-wise maximum cosine similarity derived from all models and layers. Additionally, we display the deviation of individual models from this peak to demonstrate the consistency of alignment across different architectures.

**Brain-Model RSA Analysis**. We further investigate whether the models preserve the topological structure of the ventral visual pathway using RSA. Specifically, we first identify the model features (Neurons, CLS-trained SAEs, and Patch-trained SAEs) that exhibit the highest correlation with the fMRI signal for each voxel. Both the brain signals and their corresponding best-matching model features are then reordered according to the ROIs. Finally, we employ RSA to quantitatively measure the structural similarity between these aligned representations, yielding a peak RSA score of 0.686 for Dinov2, with results for other models in Table 1. We visualize the RSM for the brain and ViT-B/16$_{CLIP}$ patch-trained SAEs in Figure 3B, demonstrating that the models preserves the structure of the brain.

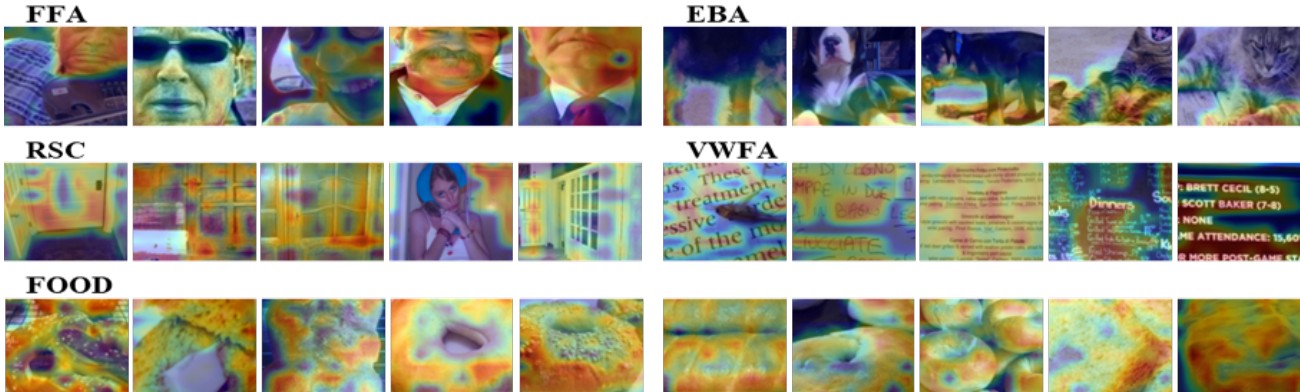

*Figure 5.* Visualization of SAEs feature activations for ViT-B/16$_{\text{CLIP}}$, S5. The heatmap utilizes a blue-to-red color scale to represent a min-max normalized activation values ranging from 0 to 1. The displayed feature was selected from the layer exhibiting the highest count of stable and functionally aligned features and we identify top-10 activated images for Food concept and top-5 images for the rest four concepts. As demonstrated, the features retain the functional selectivity consistent with the target ROI.

*Table 2.* Functional Aligned Feature Ratio. For each ROI, we identify the layer containing the highest number of stable and functionally aligned features. Within this peak layer, we calculate the proportion of these aligned features relative to the total count of stable features. A high ratio indicates that the stable representations are mainly consistent with biological functional priors.

| REGION | CLIP | DINOV2 | MAE | IMAGENET |
|--------|------|--------|-----|----------|
| FFA | 0.61 | 0.85 | 0.73 | 0.39 |
| EBA | 0.42 | 0.36 | 0.21 | 0.67 |
| RSC | 0.46 | 0.36 | 0.09 | 0.61 |
| VWFA | 0.78 | 0.56 | 1.00 | 0.72 |
| FOOD | 0.43 | 0.50 | 0.23 | 0.92 |

### 4.3. Evaluating Model Functions Identified by ROIs

Building upon the alignment established in Section 4.2, this section demonstrates that human brain activity can serve as a functional probe for interpreting deep learning models, specifically by leveraging patch-trained SAE features. Our analysis proceeds in three steps: (1) selecting top features per layer based on correlation with ROI activations; (2) validating the semantic stability and functional consistency of these features via CLIP-based text prompting and ablation analysis; and (3) providing qualitative visualizations. Collectively, these results confirm the efficacy of brain priors in interpreting model functions.

For each layer, we identify the top-100 features exhibiting the highest activation correlation with each ROI in the high-level visual cortex. We then validate the semantic stability and functional consistency of these features using CLIP and text prompts across six categories (faces, bodies, scenes, words, food, others), as detailed in Section 3.2. Specifically, we retrieve the top 20 images from the ImageNet-1k

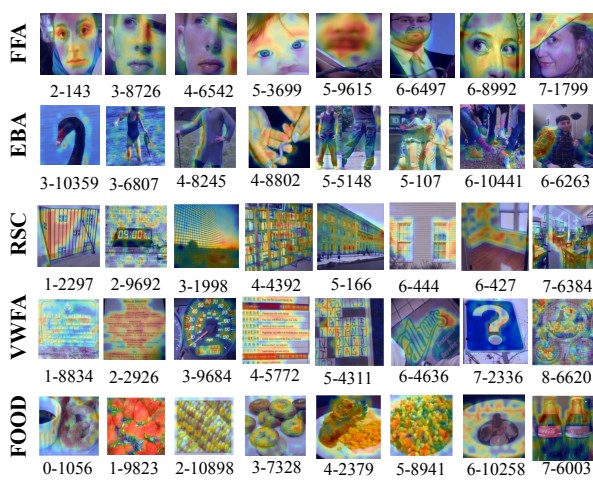

*Figure 6.* Visualization of features selected by the five target ROIs. For clarity, only one representative image is displayed per feature, annotated with its layer number and feature ID below.

test set (Russakovsky et al., 2015) that elicit the maximum activation for each selected feature. We posit that for a stable feature, at least half of these top images should belong to the same semantic category. Consequently, a feature is defined as stable and functionally aligned if its stability score $S$ exceeds 0.5. To assess the functional importance of these features, we performed an ablation analysis on SAEs reconstruction across five image categories by removing the identified stable and functionally aligned features. Figure 4 visualizes the results of the ablation—specifically the maximum impact on normalized MSE loss—alongside the count of aligned features and their proportion within the stable feature pool across all layers. Furthermore, Table 2

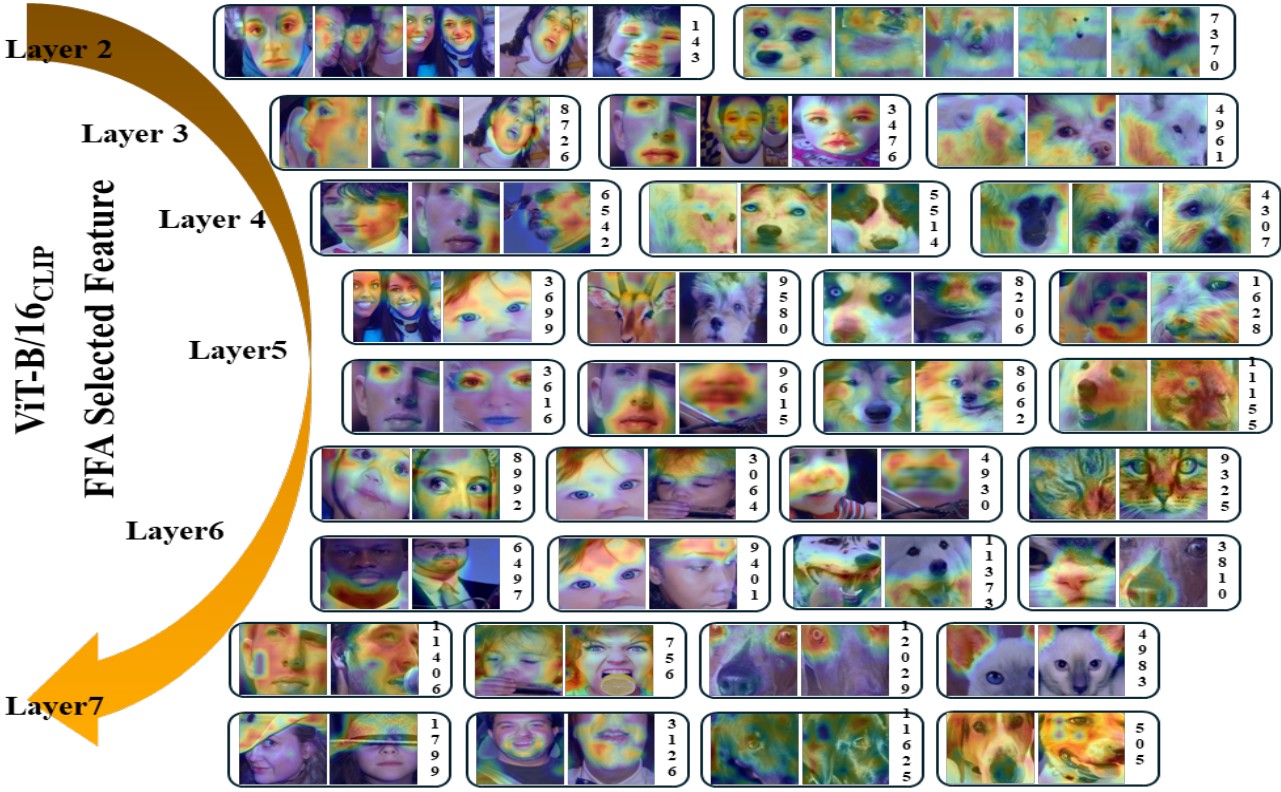

*Figure 7.* Visualization of stable and functionally aligned features for the FFA in ViT-B/16$_{CLIP}$, S5. The left part indicates the model layers (2-7), while the right part displays the corresponding features' visualization for each layer, annotated with their feature id. To conserve space, the visualization is restricted to the first 8 layers, and some redundant features in layers 4-7 (those already present in earlier layers) are omitted. This progression illustrates the trajectory of how the face concept emerges within a deep learning model.

presents the Functional Aligned Feature Ratio for the top-performing layers, where higher values signify a stronger correspondence between the selected features and the brain's functional priors. Complementing these metrics, Figure 5 provides qualitative visualizations of features selected from the layer with the highest count of functionally aligned units for ViT-B/16$_{CLIP}$, S5. These combined experiments confirm that features selected via ROI fMRI signals accurately reflect the functional priors of the corresponding brain regions. The ablation experiments are detailed in Appendix G.

### 4.4. Layer-wise Emergence of Concept Representations

Leveraging the alignment between high-level visual cortex and SAEs, this chapter traces the hierarchical emergence of five generic concepts: "Faces," "Bodies," "Places," "Words," and "Food." Figure 7 details the "Faces" concept in ViT-B/16$_{CLIP}$, while Figure 6 presents simplified visualizations for the others. Due to space constraints, we restrict visualization to the first 8 layers.

**Faces**: Notably, Face concept in ViT-B/16$_{CLIP}$ emerge unex-

pectedly early at Layer 2, contrasting with prior findings that associate these layers with low-level features (Dorszewski et al., 2025; Yang et al., 2024). By Layer 3, the model localizes human faces holistically. Interestingly, the model distinguishes humans from animals: it identifies specific facial components for humans but only surface for animals in early layers. From Layers 4 to 6, the model decomposes facial representations, with features for human "eyes", "mouth", "forehead", "neck", et.al. emerging progressively. These concepts also emerge for animals, where they gradually focus on specific animal categories. Beyond Layer 7, abstract attributes such as "hats" and specific face categories appear.

**Other Concepts**: Bodies evolve from "edges" (Layer 3) to whole-body and skin localization (Layers 4-5), finally separating into fine-grained body parts (Layer 6+); Places started from low-level patterns (lines, grids in Layers 1 and 2) to complex scenes in Layers 4-5, eventually extracting semantic context; Words emerge early in Layer 1, then gradually distinguishes numbers and different languages (Layers 2-5), finally encoding abstract semantics like signs

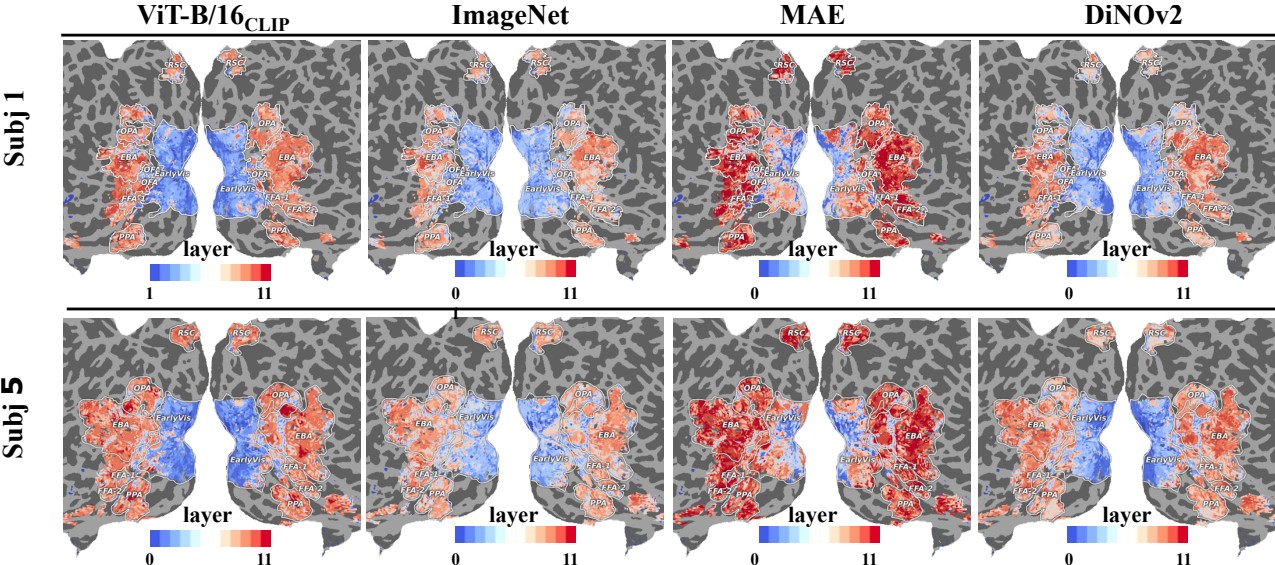

*Figure 8.* Layer-wise cortical alignment maps. For each voxel, the value represents the layer ID containing the feature with the highest correlation to that voxel's activation. We visualize these mappings for all four models on the specific cortical surfaces of Subjects 1 and 5.

and decorative fonts; Food originate from color and shape in layer 0-2, localizing food objects by Layers 4-5, finally learning semantic information such as plates and bottles in Layer 6+. The result is visualized in Figure 6.

While trajectories vary, ViT-B/16$_{CLIP}$ exhibits a general pattern: Layers 0-1 process low-level information; Layers 2-5 drive the emergence of high-level concepts; and Layer 6 onward capture abstract semantics and specific details. Moreover, different models have different recognition procedure as we discuss in Appendix I.

### 4.5. Mapping Model Layers to Brain Cortex

To dissect the functional hierarchy, we construct a voxel-wise layer mapping, assigning each voxel to the model layer with the highest activation correlations. Figure 8 not only reveals the global topography of the model's visual processing but also quantifies the cross-subject consistency. While both ViT-B/16$_{CLIP}$ and DINOv2 synchronize at Layer 9 mainly for the high-level semantic process, they differ in early visual processing. CLIP diverges from low-level features by Layer 2, whereas DINOv2 maintains early visual alignment up to Layer 4. This indicates that DINOv2 employs a more extended processing pathway for low-level features compared to the supervision-trained CLIP. Moreover, the procedure of MAEs is much later than other modes, which is consistent with its predictive training method. The reliability of the mapping is validated by a high cross-subject consistency (Average 0.89 layer-distribution similarity between S1 and S5 for ViT-B/16$_{CLIP}$). Table 3 presents the

*Table 3.* Layer Mapping Alignment. For each ROI, we calculate the layer distribution correlations between every pair of subjects. The table reports the minimum and maximum correlation values observed for each model across all subject pairs.

| REGION | CLIP | | DINOv2 | | MAE | | IMAGENET | |
|---|---|---|---|---|---|---|---|---|
| | MIN | MAX | MIN | MAX | MIN | MAX | MIN | MAX |
| V1 | 0.85 | 1.00 | 0.80 | 0.98 | 0.58 | 0.96 | 0.91 | 1.00 |
| V2 | 0.93 | 0.99 | 0.85 | 0.99 | 0.51 | 0.97 | 0.82 | 1.00 |
| V3 | 0.93 | 1.00 | 0.92 | 1.00 | 0.87 | 0.98 | 0.81 | 0.99 |
| HV4 | 0.83 | 0.99 | 0.66 | 0.97 | 0.66 | 0.97 | 0.66 | 0.97 |
| FFA | 0.72 | 0.99 | 0.87 | 0.99 | 0.95 | 1.00 | 0.81 | 0.99 |
| EBA | 0.89 | 1.00 | 0.93 | 1.00 | 0.99 | 1.00 | 0.89 | 1.00 |
| RSC | 0.68 | 0.99 | 0.92 | 0.99 | 0.96 | 1.00 | 0.96 | 1.00 |
| VWFA | 0.83 | 0.99 | 0.80 | 1.00 | 0.94 | 1.00 | 0.80 | 0.99 |
| FOOD | 0.80 | 0.99 | 0.56 | 0.97 | 0.82 | 0.99 | 0.67 | 0.99 |

maximum and minimum correlation values for each ROI distribution across all eight subjects for each model. We utilize cosine similarity to compute the correlation. The detailed analysis is presented in Appendix J.

## 5. Conclusion

In this study, we propose SAEs-BrainMap, demonstrating that the brain can serve as an effective functional probe at the ROI level for interpreting deep learning models. We first demonstrate a strong direct alignment between the model and the brain activationally and structurally. Specifically, we find that SAE features exhibit higher correlations than raw neurons. Building on this, we validate the feasibility of employing brain activation as a probe to discover functionally aligned features within the model's latent space. Our

findings not only provide a fine-grained view of generic concept emergence in vision models but also establish a robust methodology for bridging biological and artificial intelligence.

## Acknowledgements

This work was supported by the National Natural Science Foundation of China (grants number 82302175 and 62336002); the National Science and Technology Innovation 2030 Program (grant number 2021ZD0200500 and 2021ZD0200506); the Beijing Natural Science Fundation (grant number QY24183). We thank the National Center for Protein Sciences at Peking University in Beijing, China for assistance on data analysis. This work was partially supported by Beijing Institute of Technology Kunpeng&Ascend Center of Cultivation.

## Impact Statement

This paper presents work whose goal is to advance the field of Deep Learning Interpretability. There are many potential societal consequences of our work, none which we feel must be specifically highlighted here.

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

# A. Limitations and Future Works

In this work, we introduced the SAE-BrainMap framework to facilitate a layer-wise analysis of model functionality. However, our current experimental approach entails certain limitations, which we discuss below alongside directions for future research.

First, our experiments are currently confined to the Vanilla SAEs introduced by (Huben et al., 2024). We have not yet extended our analysis to incorporate more recent variants, such as TopK SAEs (Gao et al., 2025) or Archetypal SAEs (Fel et al., 2025), which may improve the brain-model alignment.

Second, we employed a relatively coarse-grained method to evaluate SAEs feature functionality. While this approach establishes consistency between the dominant semantics of input images and brain ROI selectivity, it does not guarantee a perfect one-to-one mapping between SAEs feature selectivity and primary image semantics. Nevertheless, we argue that this methodology is sufficient for our purposes. Primarily, our features are selected based on actual activations rather than a certain vector in the latent space. Furthermore, we implemented a stability threshold based on the premise that if a feature aligns with a specific ROI, the majority of images triggering maximal activation should share semantics consistent with that ROI's function. This threshold effectively filters for functional consistency, and while some features may inevitably be overlooked, the approach demonstrates strong empirical performance.

Third, we restricted our analysis to the top-100 most correlated features per ROI. While this cutoff may exclude some relevant units, particularly in deeper layers, it was chosen for two reasons: it aligns with the protocol in (Thasarathan et al., 2025), and it represents approximately 1% of the total features, providing a sufficient sample for statistical analysis. Additionally, we characterized feature functionality using only the top-20 maximally activating images. We acknowledge that due to superposition (Elhage et al., 2022), where features may still activate for semantically related concepts at lower intensities even if the feature is captured by SAEs and expected to be sparse. This method might not capture the full polysemantic nature of a feature. We use this evaluation strategy follows established precedents set by (Thasarathan et al., 2025; Fel et al., 2025).

What's more, although SAEs-BrainMap provides a neuroscience-informed way to identify model features aligned with human visual functions, the concepts that can be interpreted by our framework are inherently constrained by the knowledge of the human visual system.

In future work, we aim to deepen our investigation into the layer-wise analysis pipeline and extend our experiments to include quantitative cross-model similarity analyses. While we currently rely on visualization to demonstrate functional differences between architectures, we plan to develop rigorous metrics to quantify these divergences systematically.

# B. Comparison with Text-based and Neuron-based Feature Discovery

To further examine whether brain-guided feature selection provides additional benefits over other interpretation methods, we conduct supplementary comparisons with two text-based feature dissection baselines. The first baseline is **CLIP-Dissect** (Oikarinen & Weng, 2022), which interprets raw model neurons by matching their activation patterns with CLIP-based text prompts. The second baseline is **CLIP-SAEs**, which follows the same procedure as CLIP-Dissect but applies the text-based matching process to SAE features instead of raw neurons. These two baselines allow us to evaluate whether the proposed brain-guided probing strategy is necessary, or whether comparable concept localization can be achieved by directly using CLIP text embeddings.

Specifically, given a set of images $\mathcal{X} = \{x_i\}_{i=1}^{n}$ and a vocabulary of text prompts $\mathcal{T} = \{t_j\}_{j=1}^{m}$, we first compute the CLIP image-text similarity matrix

$$M_{ij} = \cos\left(E_{\text{img}}(x_i), E_{\text{text}}(t_j)\right), \quad M \in \mathbb{R}^{n \times m}, \tag{5}$$

where $E_{\text{img}}(\cdot)$ and $E_{\text{text}}(\cdot)$ denote the CLIP image and text encoders, respectively. For CLIP-Dissect, we collect the activation matrix of raw model neurons over the same image set, denoted as $S \in \mathbb{R}^{n \times k}$, where $k$ is the number of neurons or activation dimensions. Each neuron is then assigned to the text prompt whose CLIP similarity vector has the highest correlation with its activation vector. We use $f_a$ to denotes the function of neuron $a$:

$$\text{score}(f_a, t_j) = \text{corr}\left(S_{:,a}, M_{:,j}\right), \quad \hat{t}_a = \arg\max_{t_j \in \mathcal{T}} \text{score}(f_a, t_j). \tag{6}$$

CLIP-SAEs uses the same scoring rule, but replaces the raw neuron activation matrix $S$ with the sparse activation matrix of

## Face concept extraction comparation for CLIP ViT-B/16

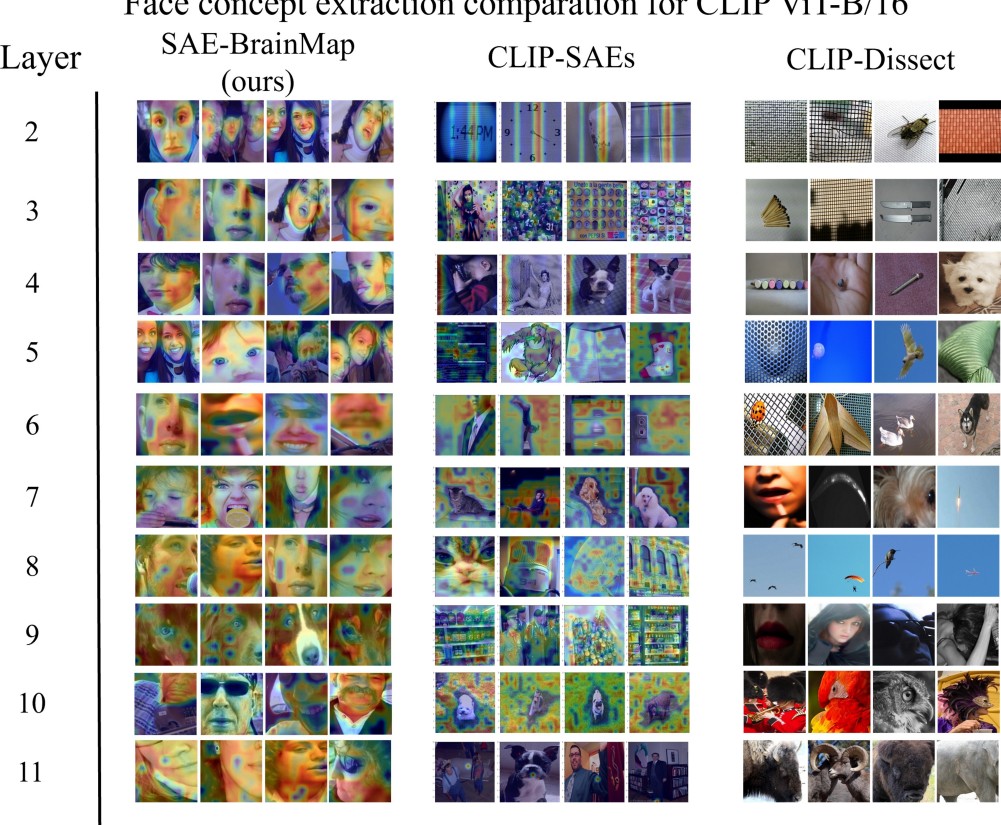

*Figure 9.* Result visualization between our method and two baseline.

SAE features. We then compare the selected ViT-B/16$_{CLIP}$ features from different methods to evaluate the concept selectivity performance and the result is show in Figure 9.

The comparison shows that brain-guided feature selection identifies face-related selective features at earlier layers and with stronger semantic coherence than both CLIP-Dissect and CLIP-SAEs. This suggests that although CLIP-based text prompts provide a useful semantic reference, they may still be biased toward language-aligned visual concepts and may not fully capture the functional organization of model features. In contrast, using category-selective brain regions as external functional probes provides a complementary source of supervision, enabling SAEs-BrainMap to locate features that are not only semantically meaningful but also functionally aligned with human visual processing.

# C. Experiments Details

The fMRI dataset we use in the experiment is Nature Scenes Dataset (NSD) (Allen et al., 2022) which includes high-density fMRI data from eight participants. Each of the subject was expected to view 10,000 natural scene images selected from COCO (Lin et al., 2014) for three times, while all the subject share the same set of 1000 images and an independent set of 9000 images. fMRI beta values were z-scored across runs and averaged across up to three repetitions per image, yielding one fMRI response per voxel per image. Four of the subjects (S1,S2,S5,S7) finished all the experiment, while another four did not. We performed the same preprocessing for subjects who did not finish all the tasks as (Yang et al., 2025; Xue et al., 2024; Cerdas et al., 2024).The cortical visualization results were rendered in each subject's native space using Pycortex (Gao et al., 2015)

The experiment is performed based on Pytorch (Paszke et al., 2019) on four Nvidia L20 (48GB each). We train SAEs for each layer of each model in our analyses on CLS Token and the rest Patches separately. SAEs' feature dimension $k = R \times d$, where $d$ denotes target layer's activation dimension, and $R$ is a hyperparameter which is kept $R = 16$ for all of the models. The hyperparameter $\alpha = 0.00086$, as (Huben et al., 2024). We train all SAEs with the ImageNet-1k 2012 training set (Russakovsky et al., 2015) for 5 epochs and learning rate set to $5 \times 10^{-5}$. The pretrained vision models used in our experiment are OpenAI CLIP visual backbone ViT-16/B (Radford et al., 2021), MAE ViT-B/16 (He et al., 2022), DiNOv2 ViT-B/14 (Oquab et al., 2023), and ImageNet trained ViT-16/B (Dosovitskiy et al., 2020)(the model is named ImageNet in the paper) loaded from pytorch (Paszke et al., 2019).

# D. CLIP Evaluation Prompt

We employed the following text prompts and the CLIP model to classify the feature-selective images.Our prompts are mainly based on (Yu et al., 2026). The specific prompts are detailed below:

**Faces** [A photo of a person's face, A portrait photo of a face, A face facing the camera, A photo of a face, A photo of an animal's face, A photo of faces, People looking at the camera, A portrait of a person, A portrait photo]

**Bodies** [A photo of a torso, A photo of limbs, A photo of bodies, A photo of people, A photo of animals, A photo of a body, A person's arms, A person's legs, A photo of hands]

**Places** [A photo of a bedroom, A photo of an office, A photo of a hallway, A photo of a doorway, A photo of interior design, A photo of a building, A photo of a house, A photo of nature, A photo of a landscape]

**Food** [A photo of food, A photo of cuisine, A photo of fruit, A photo of foodstuffs, A photo of a meal, A photo of bread, A photo of rice, A photo of a snack, A photo of pastries]

**Words** [A photo of words, A photo of glyphs, A photo of a glyph, A photo of text, A photo of numbers, A photo of a letter, A photo of letters, A photo of writing, A photo of text on an object]

**Others** [A photo of a single solid color, A photo emphasizing smooth color gradients, A photo emphasizing high contrast, A photo of texture, A photo containing simple geometric shapes only, A photo emphasizing edges and contours, A photo of grid-like pattern, A photo with aligned elements, A blurred photo]

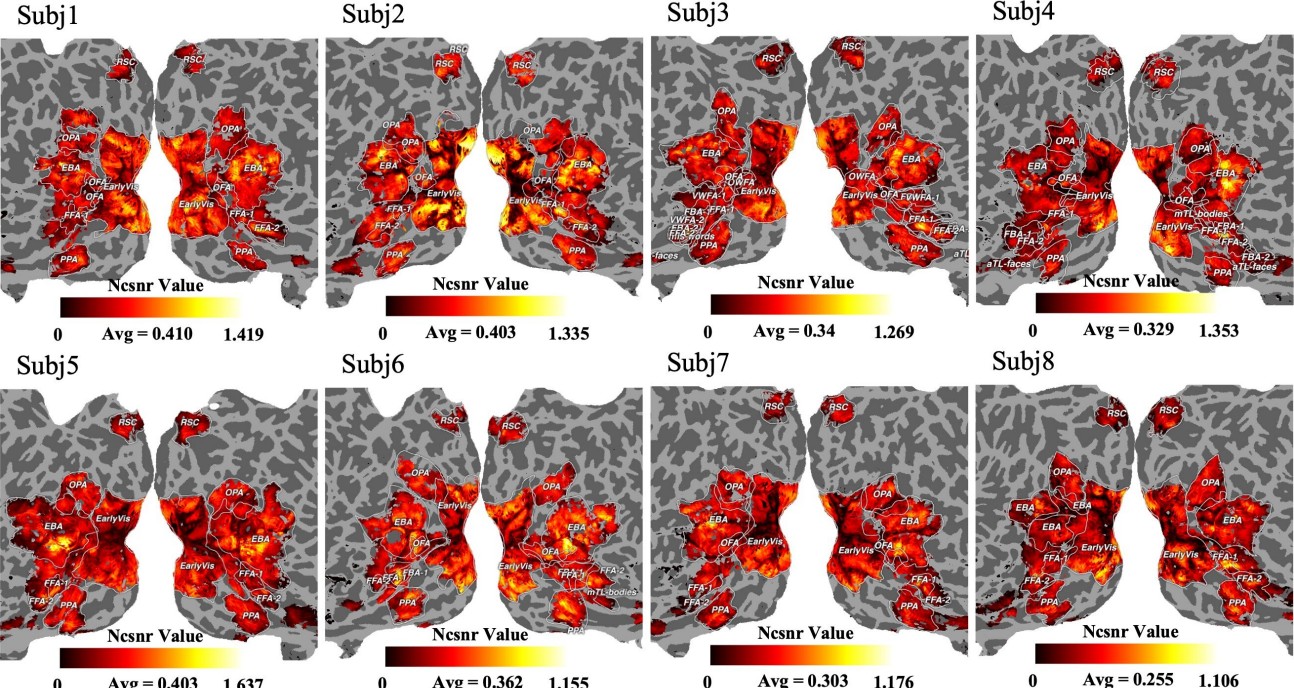

*Figure 10.* The visualization of the ncsnr value for Subject 1-8 in the ventral visual pathway. The ncsnr is defined as the ratio of a voxel's 'signal standard deviation' to its 'noise standard deviation'.

## E. Signal-to-Noise Ratio of fMRI Data and Evaluation of Activation Similarity

In the preceding section, we stated that the average activation correlation between ViT-B/16$_{\text{CLIP}}$ and the brain reached 0.28, and the RSA score reached 0.596, with DINOv2 achieving as high as 0.686. This represents a strong correlation, a conclusion we support based on the dataset paper (Allen et al., 2022). Using the NSD dataset, we can derive the ncsnr of the fMRI data. The ncsnr is defined as the ratio of a voxel's 'signal standard deviation' to its 'noise standard deviation'. Furthermore, the ncsnr can be used to evaluate the noise ceiling of the fMRI data, which represents the maximum percentage of signal that can be theoretically explained. The formula is given by:

$$ncsnr = \frac{\sigma_{signal}}{\sigma_{noise}} \qquad NC = 100 \times \frac{ncsnr^2}{ncsnr^2 + \frac{1}{n}}$$

where $n$ is the average number of trials. For Subject 5, the average ncsnr in the ventral visual pathway is 0.40, with a maximum ncsnr of 1.64, corresponding to an average noise ceiling of 36%. As noted in the experimental paper, in terms of Pearson's correlation, **"this is equivalent to a prediction accuracy of** $r = 0.60$**"** (Allen et al., 2022). Our experimental correlations are of the same order of magnitude as this benchmark. The visualization results for the ncsnr of Subject 5 are shown in the Figure 10.

Furthermore, we further calculated the Brain Encoder as a reference to demonstrate that the current correlation is a relatively high value. We train brain encoders for each layer and each model. For each model, we extract j-th layer's activation $A^{(j)}$ and training a linear layer with equation: $A^{(j)}W + b = B$, $B$ is the voxel activation for a certain image. We use AdamW Optimizer through the training process. We use the independent 9000 image for each subject as (Luo et al., 2023). We performed image preprocess as (Luo et al., 2024), input images are resized to $224 \times 224$ pixels. Data augmentation includes randomly scaling pixel intensities within the range [0.95, 1.05], followed by normalization using the mean and standard deviation of CLIP-preprocessed images. Prior to being fed into the network, each image undergoes a random spatial offset of up to 4 pixels along both the horizontal and vertical axes, with edge padding used to fill any resulting empty regions. Additionally, independent Gaussian noise with mean $\mu = 0$ and variance $\sigma^2 = 0.05$ is added to each pixel. We measure the

predict performance with *cosine similarity* and $R^2 score$, illustrating in Table 4.

| SUBJECT | ALL MODEL | | VIT-B/16$_\text{CLIP}$ | | IMAGENET | | MAE | | DINOv2 | |
|---|---|---|---|---|---|---|---|---|---|---|
| *Cos Similarity* | MAX | AVG | MAX | AVG | MAX | AVG | MAX | AVG | MAX | AVG |
| S1 | 0.868 | 0.372 | 0.868 | 0.306 | 0.838 | 0.287 | 0.751 | 0.245 | 0.859 | 0.289 |
| S2 | 0.868 | 0.368 | 0.868 | 0.307 | 0.827 | 0.289 | 0.799 | 0.250 | 0.862 | 0.296 |
| S3 | 0.848 | 0.323 | 0.842 | 0.265 | 0.798 | 0.249 | 0.706 | 0.214 | 0.825 | 0.254 |
| S4 | 0.856 | 0.319 | 0.856 | 0.258 | 0.819 | 0.242 | 0.721 | 0.208 | 0.846 | 0.248 |
| S5 | 0.889 | 0.414 | 0.889 | 0.338 | 0.848 | 0.318 | 0.799 | 0.274 | 0.880 | 0.324 |
| S6 | 0.812 | 0.324 | 0.802 | 0.259 | 0.778 | 0.244 | 0.700 | 0.209 | 0.798 | 0.248 |
| S7 | 0.870 | 0.327 | 0.862 | 0.264 | 0.840 | 0.246 | 0.759 | 0.212 | 0.864 | 0.253 |
| S8 | 0.791 | 0.252 | 0.783 | 0.206 | 0.723 | 0.196 | 0.685 | 0.171 | 0.769 | 0.199 |
| **COS AVG** | 0.850 | 0.337 | 0.846 | 0.275 | 0.809 | 0.259 | 0.740 | 0.223 | 0.838 | 0.264 |

*Table 4.* **Brain Encoder Cos Similarity**. The cosine similarity between Brain Encoder Predictions and the voxel fMRI response. Our findings reveal a strong intrinsic alignment between the model and the brain, with direct activation correlations being comparable to predicted activation correlations.

# F. Brain-Model similarity for all subjects

In this section, we visualize the model-brain correlations for each subject. The results for subject 1-8 is presented in Figure 11.

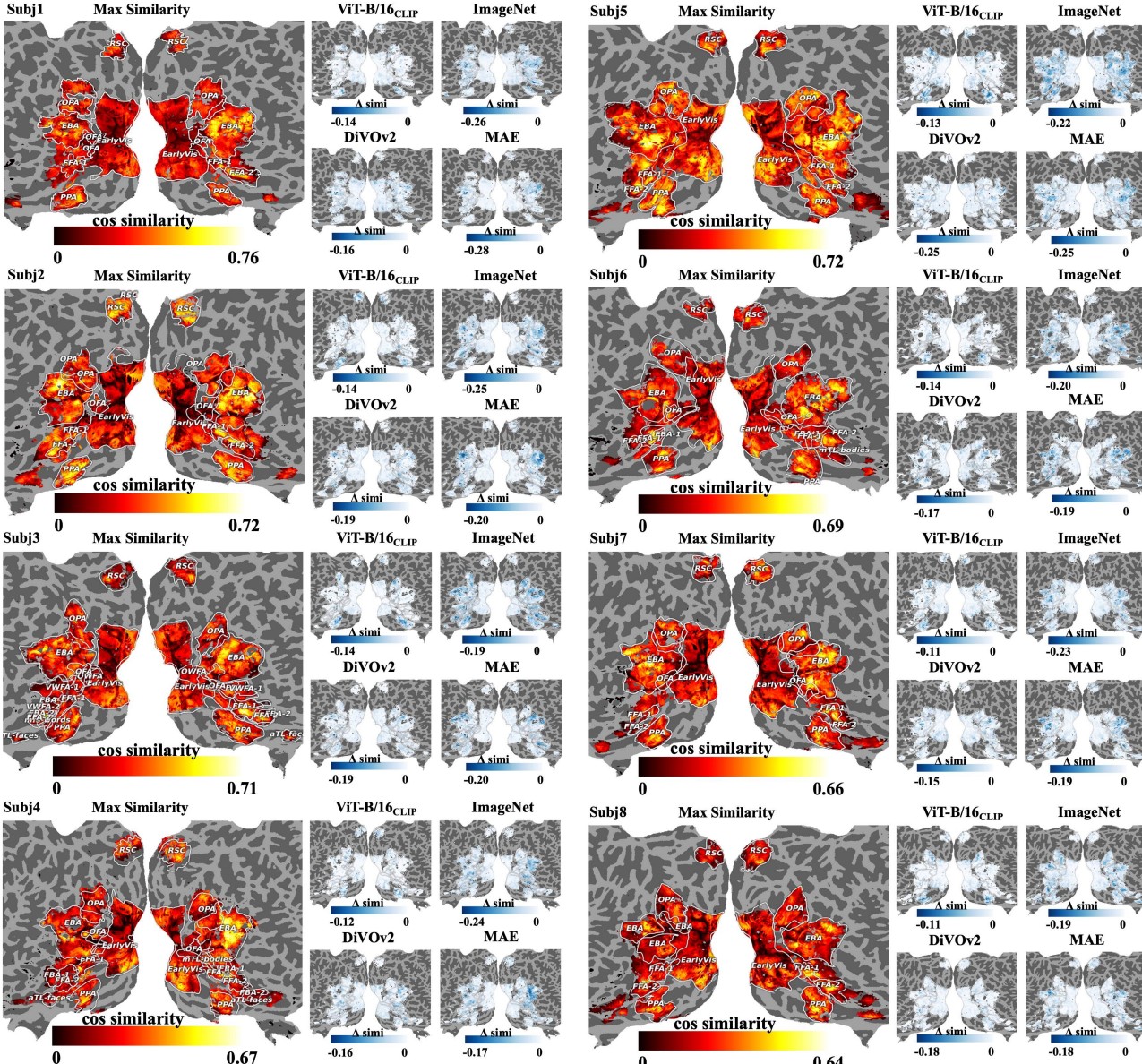

*Figure 11.* Model-Brain Similarity. Here, we visualize the model-brain cosine correlations for eight subjects across four models.

## G. Details of Ablation Analysis

To further validate the ROI-relevance of the selected features, we conducted ablation studies. As the procedural details were omitted from the main text due to space constraints, we elaborate on the experimental methodology here. For the ablation analysis, we utilized the 'shared' images from the NSD dataset. Given that some subjects did not complete the full experimental protocol, we excluded images that were not viewed by the complete set of subjects, ensuring a common image set. Following the design of the category functional localizer (fLoc) experiments (Stigliani et al., 2015), we first classified these images into specific subcategories: 'adult', 'body', 'car', 'child', 'corridor', 'house', 'instrument', 'limb', 'number' and 'word'. These were subsequently aggregated into five major categories—'Faces', 'Bodies', 'Places', 'Words', and 'Food'—to facilitate the ablation experiments. For the ablation method, we permanently set the activations of the selected features(those features labeled as functionally aligned and stable by our methods) to zero. To validate the effectiveness of the ablation experiments, we focused on the Normalized MSE Loss of the SAE reconstruction, which can be formulated as:$\mathcal{L}_{\mathrm{NMSE}} = \frac{\|\mathbf{x}-\hat{\mathbf{x}}\|_2^2}{\|\mathbf{x}\|_2^2}$ where $\mathbf{x}$ represents the original input and $\hat{\mathbf{x}}$ denotes the reconstructed output. The Layer-wise experiment result are presented in Table 5.

*Table 5.* Ablation study results: Reconstruction Normalized MSE Loss across ROIs selected Features (Rows) and stimulus sub-categories (Columns).

| ROI | ADULT | BODY | CAR | CHILD | CORR. | FOOD | HOUSE | INSTR. | LIMB | NUM. | WORD |
|---|---|---|---|---|---|---|---|---|---|---|---|
| FACES | 0.4100 | 0.3425 | 0.1588 | **0.4920** | 0.1400 | 0.1640 | 0.1041 | 0.3292 | 0.2736 | 0.1961 | 0.1265 |
| BODIES | 0.2499 | 0.2326 | 0.1905 | 0.2036 | 0.1540 | 0.1691 | 0.1579 | **0.2742** | 0.2015 | 0.1860 | 0.1655 |
| PLACES | 0.1825 | 0.1536 | 0.2148 | 0.2134 | **0.3559** | 0.1269 | 0.3183 | 0.3468 | 0.1203 | 0.1416 | 0.1606 |
| FOOD | 0.1406 | 0.1453 | 0.1527 | 0.1515 | 0.1384 | **0.3509** | 0.1174 | 0.1589 | 0.1521 | 0.1558 | 0.1526 |
| WORDS | 0.1787 | 0.1925 | 0.3679 | 0.1882 | 0.1753 | 0.2274 | 0.1927 | 0.1631 | 0.2047 | 0.3568 | **0.4209** |

## H. Evaluation on CNN structure

Our proposed method is equally applicable to the analysis of CNN architectures; however, due to inherent architectural differences, we present these analysis results in the Appendix. Since CNNs do not distinguish between [CLS] and patch tokens, we adopt the methodology described in (Oikarinen & Weng, 2022) to extract internal layer-wise image encoding vectors. Specifically, within each layer, we compute the average across spatial units to obtain a one-dimensional vector representation for a single image. The training hyperparameters and datasets remain consistent with those used for the [CLS] SAEs, with detailed specifications provided in Section C. We visualize the correlation between ResNet50$_{\mathrm{CLIP}}$ and brain activation, along with the results of the layer-wise mapping. For the purpose of this visualization, we designate the initial three convolutional layers of ResNet50 as the model's first layer, resulting in a total of five layers.

Furthermore, our method allows for a qualitative elucidation of the differences between the two models and aligns with previous research findings. Specifically, (Kornblith et al., 2019) introduces CKA method, a kind of representation analysis method similar to Representation Similarities to explore two deep learning models' activation similarity. For model $A$ and $B$, let $K_{ij} = k(\mathbf{x}_i, \mathbf{x}_j)$, $L_{ij} = l(\mathbf{y}_i, \mathbf{y}_j)$, where $k, l$ are kernel functions and $\mathbf{x}_i$ and $\mathbf{y}_i$ are the i-th image activation of model A and model B. Define the centering matrix: $H_n = I_n - \frac{1}{n}\mathbf{1}\mathbf{1}^\top$. The CKA calculate method is discribed as follow:

$$\mathrm{HSIC}(K, L) = \frac{1}{(n-1)^2}\mathrm{tr}(KHLH). \tag{7}$$

$$\mathrm{CKA}(K, L) = \frac{\mathrm{HSIC}(K, L)}{\sqrt{\mathrm{HSIC}(K, K) \cdot \mathrm{HSIC}(L, L)}}. \tag{8}$$

We compare the CKA matrix between ViT-B/16$_{\mathrm{CLIP}}$ and ResNet50$_{\mathrm{CLIP}}$, and calculate the similarity between each model each layer's Brain-Model Layer Alignment Result, as illustrate in Figure 12. The similarity is calculated through a method similar to Intersection over Union (IoU). The similarity score is calculated through Equation: $\mathrm{IoU}_{i,j} = \frac{|\{v | l_{\mathrm{ViT}}(v)=i \wedge l_{\mathrm{ResNet}}(v)=j\}|}{|\{v | l_{\mathrm{ViT}}(v)=i \vee l_{\mathrm{ResNet}}(v)=j\}|}$ Here, $l_{\mathrm{ViT}}(v)$ denotes the index of the layer in ViT that best matches voxel $v$, and $l_{\mathrm{ResNet}}(v)$ denotes the corresponding best-matching layer in ResNet. The numerator represents the number of voxels for which both models assign the same respective layer (i.e., the intersection), while the denominator represents the number of voxels assigned to that layer by at

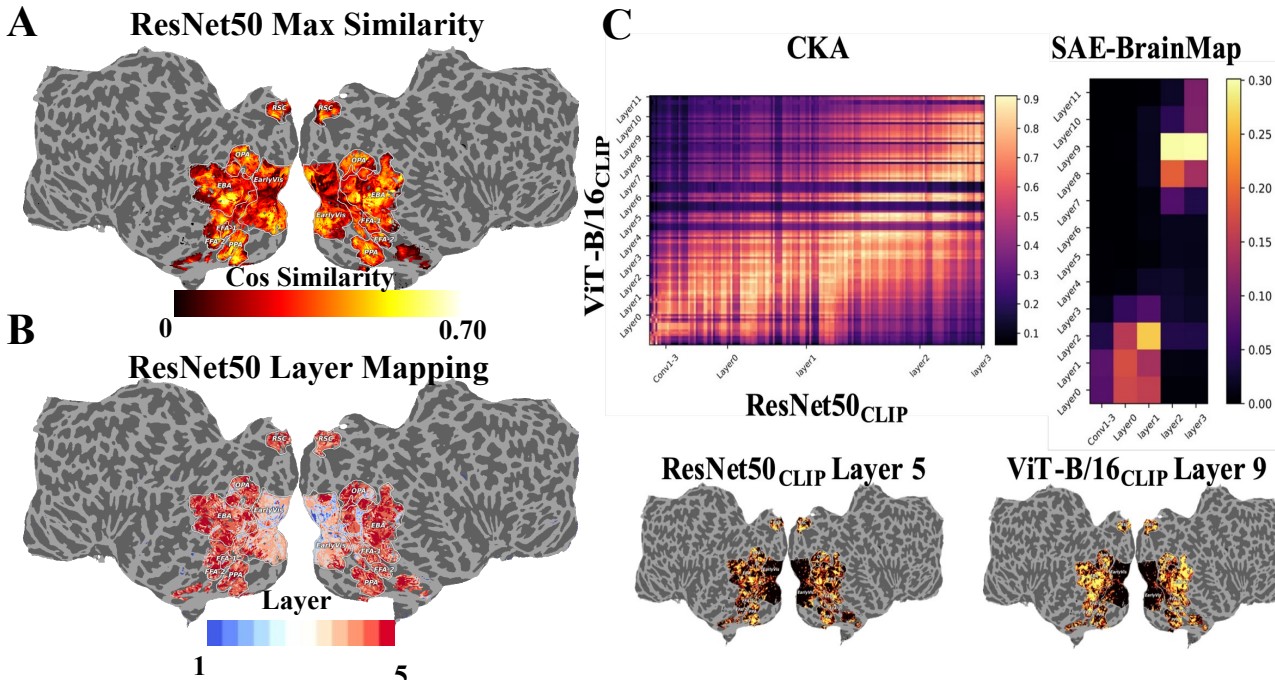

*Figure 12.* The visualization of experiments on ResNet50$_{\text{CLIP}}$. Part A visualizes the correlation distribution between ResNet50$_{\text{CLIP}}$ and brain activation. Part B illustrates the hierarchical mapping between the model and the brain. Part C qualitatively compares the hierarchical correlations between ResNet50$_{\text{CLIP}}$ and ViT-B/16$_{\text{CLIP}}$, benchmarking our findings against CKA results. Additionally, the bottom of Part C displays the distribution of the corresponding model layers across the cerebral cortex, offering a potential explanation for the CKA experimental results.

least one of the two models (i.e., the union). We found that our layer alignment IoU could reveal the activation similarity pattern between two models activation similarity, we visualize the voxel that similar to layer 9 of ViT-B/16$_{\text{CLIP}}$ and layer 5 of ResNet50$_{\text{CLIP}}$ in Figure 12.

# I. Differences between ViT-B/16$_{CLIP}$ and DINOv2 in "Face" concepts

In this section, we employ visualization techniques to conduct a preliminary investigation into the differences between ViT-B/16$_{CLIP}$ and DINOv2 in face recognition tasks. This analysis is qualitative in nature; we perform a functional analysis and elaboration based solely on the selected stable features, rather than quantitatively demonstrating the exact of these disparities. The visualization results indicate that, the "Face" concept emerging in DINOv2 is different with that in ViT-B/16$_{CLIP}$. DINOv2 emerges the "Face" later and start from the animal faces. ViT-B/16$_{CLIP}$ emerges the "Face" earlier and start from both the human faces and animal faces. This is largely due to the training method difference of DINOv2 and ViT-B/16$_{CLIP}$. The introduction of natural language enables the ViT-B/16$_{CLIP}$ to grasp high-level semantic concepts at an earlier stage. We present the visualization results in Figure 13. The result of ImageNet and MAE is present in Figure 14.

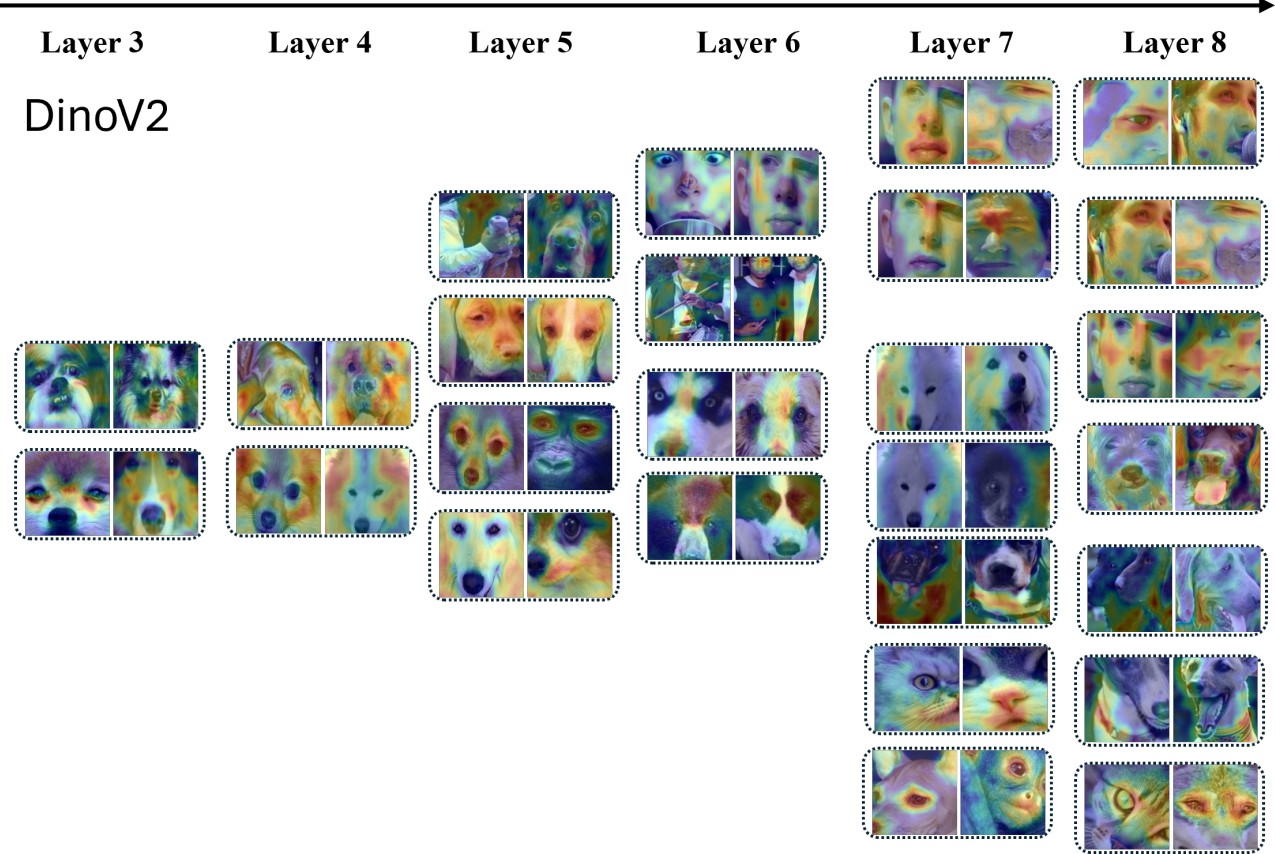

Figure 13. The visualization of the emergence of "Face" concept in DINOv2. We observe that in DINOv2, the concept of 'face' initially originates from animal features; fine-grained feature selectivity begins to emerge only at layer 5, with features specific to human faces appearing as late as layer 6. This contrasts with CLIP, where the 'face' concept emerges as early as layer 2, appearing simultaneously for both human and animal faces. This suggests that CLIP's joint image-text training approach enables the model to acquire a broader semantic understanding of 'face' at an earlier stage.

## J. Statistical Analysis of Layer Mapping

In this section, we detail the methodology used to evaluate the cross-subject consistency of the layer-wise cortical mapping. Due to inter-subject variability in the number of voxels and the spatial distribution of ROIs within the ventral visual pathway, direct voxel-to-voxel comparison is infeasible. Consequently, we adopt a coarse-grained statistical approach, complemented by qualitative visualization, to quantify consistency across subjects. For any two subjects $a$ and $b$, let $\mathcal{R}$ denote the set of common ROIs. For a given ROI $r \in \mathcal{R}$, we construct a layer distribution vector $\mathbf{d}_r \in \mathbb{R}^L$ (where $L = 12$), which captures the frequency of layer assignments within that region. The $k$-th element of this vector represents the number of voxels in ROI $r$ that are maximally correlated with the $k$-th layer of the model:

$$\mathbf{d}_r[k] = \sum_{v \in V_r} \mathbb{I}(l(v) = k), \quad k \in \{1, \ldots, 12\}$$

where $V_r$ denotes the set of voxels in ROI $r$, $l(v)$ is the layer ID assigned to voxel $v$, and $\mathbb{I}(\cdot)$ is the indicator function. The cross-subject consistency score $S(a, b)$ is then defined as the mean cosine similarity of these distribution vectors across all shared ROIs:

$$S(a, b) = \frac{1}{|\mathcal{R}|} \sum_{r \in \mathcal{R}} \frac{\mathbf{d}_r^{(a)} \cdot \mathbf{d}_r^{(b)}}{\|\mathbf{d}_r^{(a)}\| \|\mathbf{d}_r^{(b)}\|}$$

The quantitative results are presented in the Table 3. We report the minimum and maximum pairwise similarity scores across the eight subjects for all four models. Additionally, we identify the dominant layer ID—defined as the layer assigned to the plurality of voxels within a specific ROI—for each subject, characterizing the layer selectivity distribution.

*Table 6.* **Similarity and Slope**. The similarity of brain-model layer alignment between SAE-BrainMap and FactorTopy (Yang et al., 2024), as well as with the Max $R^2$ (Wang et al., 2023).

| | ViT-B/16$_{\text{CLIP}}$ | | | ImageNet | | | MAE | | | DINOv2 | | |
| --- | --- | --- | --- | --- | --- | --- | --- | --- | --- | --- | --- | --- |
| | S1 | S5 | Avg | S1 | S5 | Avg | S1 | S5 | Avg | S1 | S5 | Avg |
| Simi with FactorTopy | 0.79 | 0.54 | 0.69 | 0.91 | 0.84 | 0.70 | 0.25 | 0.20 | 0.22 | 0.54 | 0.55 | 0.61 |
| Simi with Max $R^2$ | 0.59 | 0.65 | 0.73 | 0.61 | 0.73 | 0.68 | 0.69 | 0.68 | 0.70 | 0.82 | 0.88 | 0.84 |

## K. Comparison of Different Layer Mapping Methods

Prior studies have also utilized the cerebral cortex to visualize the global information processing flow of models (Wang et al., 2023; Yang et al., 2024). However, unlike our approach, these methods mainly rely on predictive paradigms, assessing correlation via the predictive performance of Brain Encoders. This represents an indirect alignment strategy. In contrast, by directly comparing the activations of these two cognitive systems, our method mitigates potential biases introduced by training procedures and dataset characteristics. We excluded this comparison from the main text due to the absence of an absolute 'ground truth' standard for such layer mapping techniques. In this section, we visualize the results from two related works and evaluate the distributional correlation between their mappings and ours. As illustrated in the Figure 15, Figure 16, Figure 17 and Figure 18. The average results on all ROIs in Table 6. Our method exhibits a degree of distributional correlation with these prior approaches. Analysis is based on the cls-based SAEs.

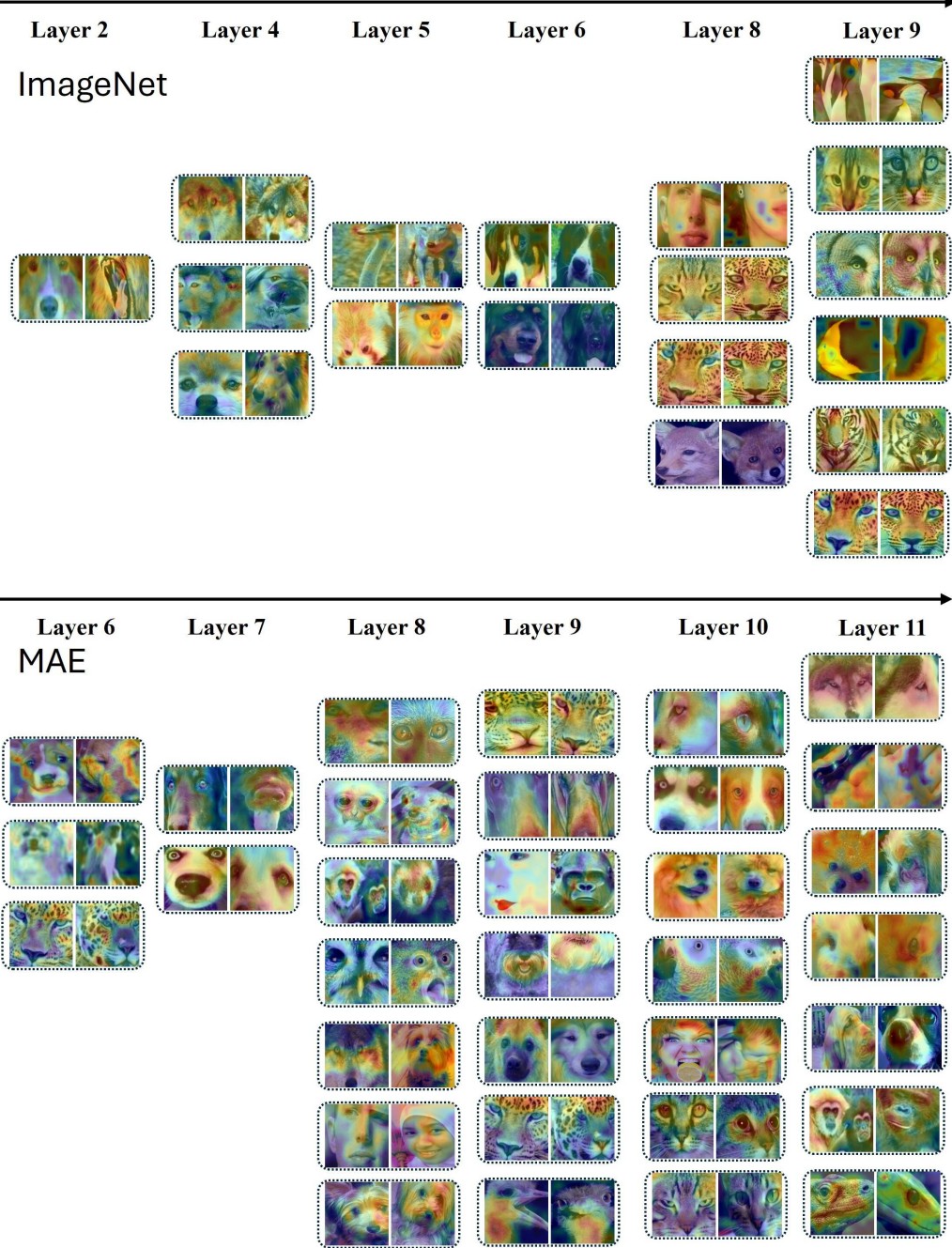

Figure 14. The visualization of the emergence of "Face" concept in ImageNet and MAE. It can be observed that the ImageNet-trained Vision Transformer struggles to emerge fine-grained semantic information regarding faces, prioritizing instead semantic concepts related to the overall category. In contrast, while MAE does exhibit fine-grained semantic information (emerging at a relatively later stage, Layer 6), its overall cognitive capability is inferior to that of the other two models.

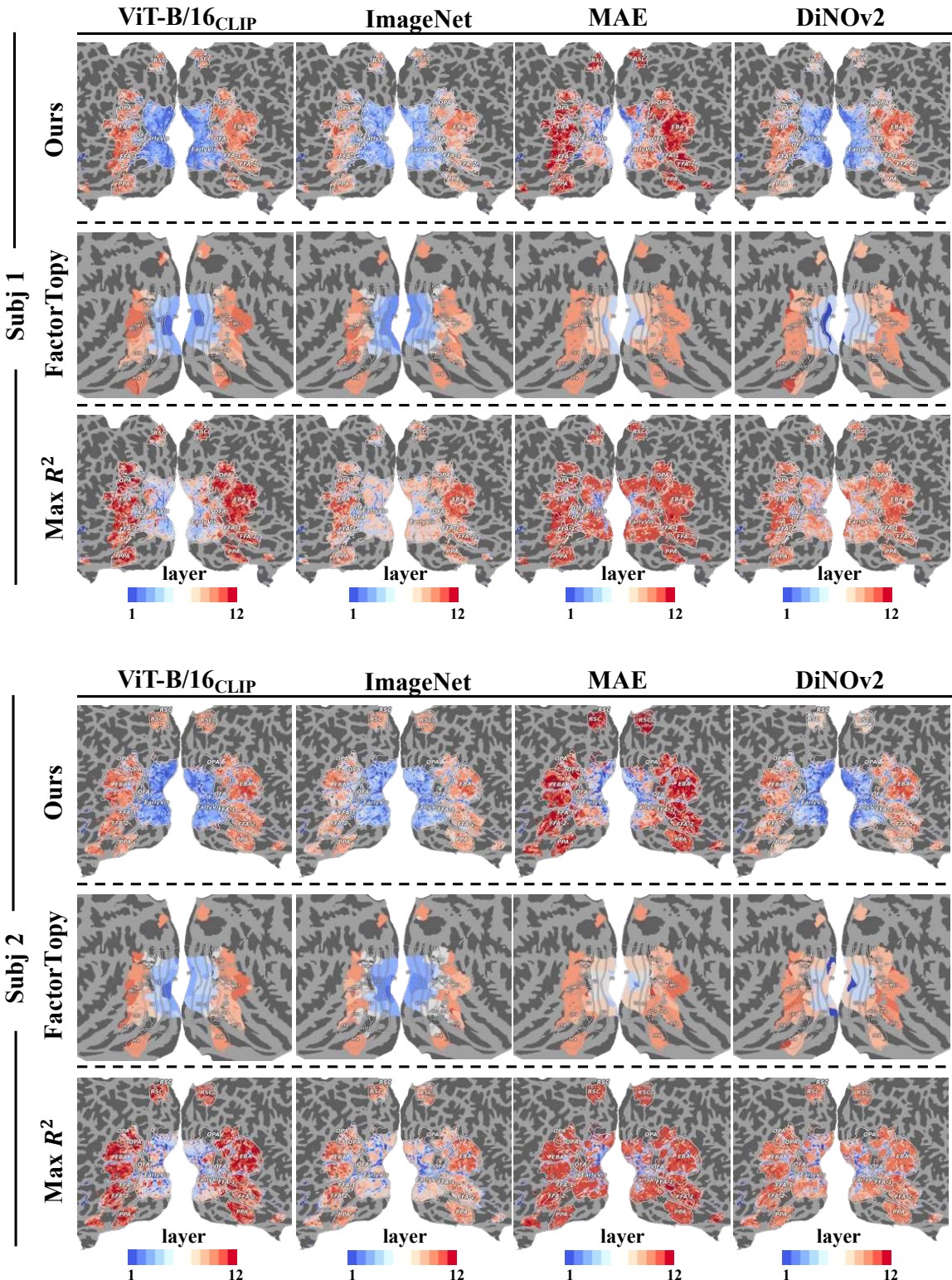

*Figure 15.* Top: Layer alignment for S1,S2 based on SAE-BrainMap, FactorTopy (Yang et al., 2024) and Max $R^2$ (Wang et al., 2023), we visualize five models with attention structure and have 12 layers.

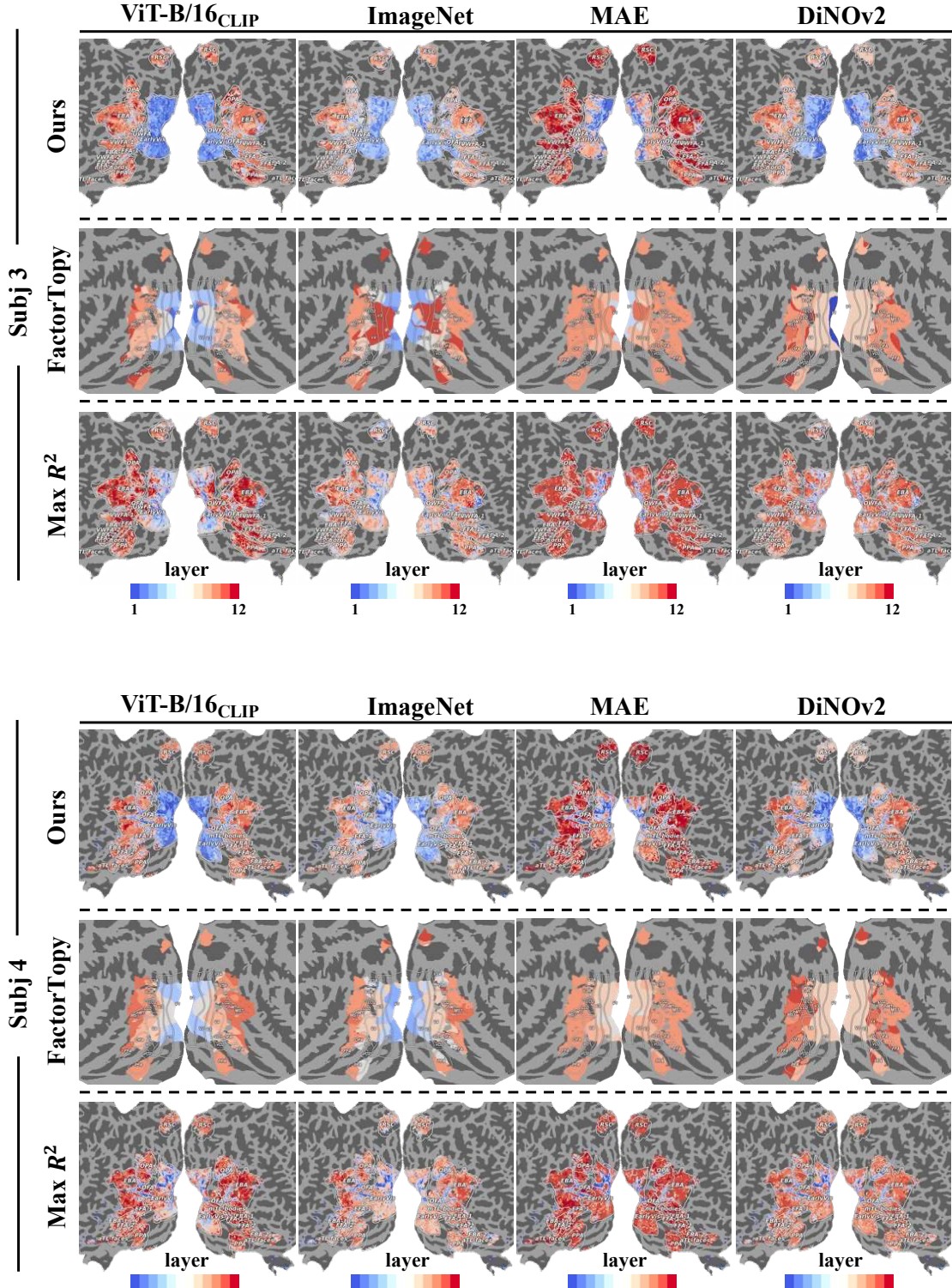

*Figure 16.* Top: Layer alignment for S3,S4 based on SAE-BrainMap, FactorTopy (Yang et al., 2024) and Max $R^2$ (Wang et al., 2023), we visualize five models with attention structure and have 12 layers.

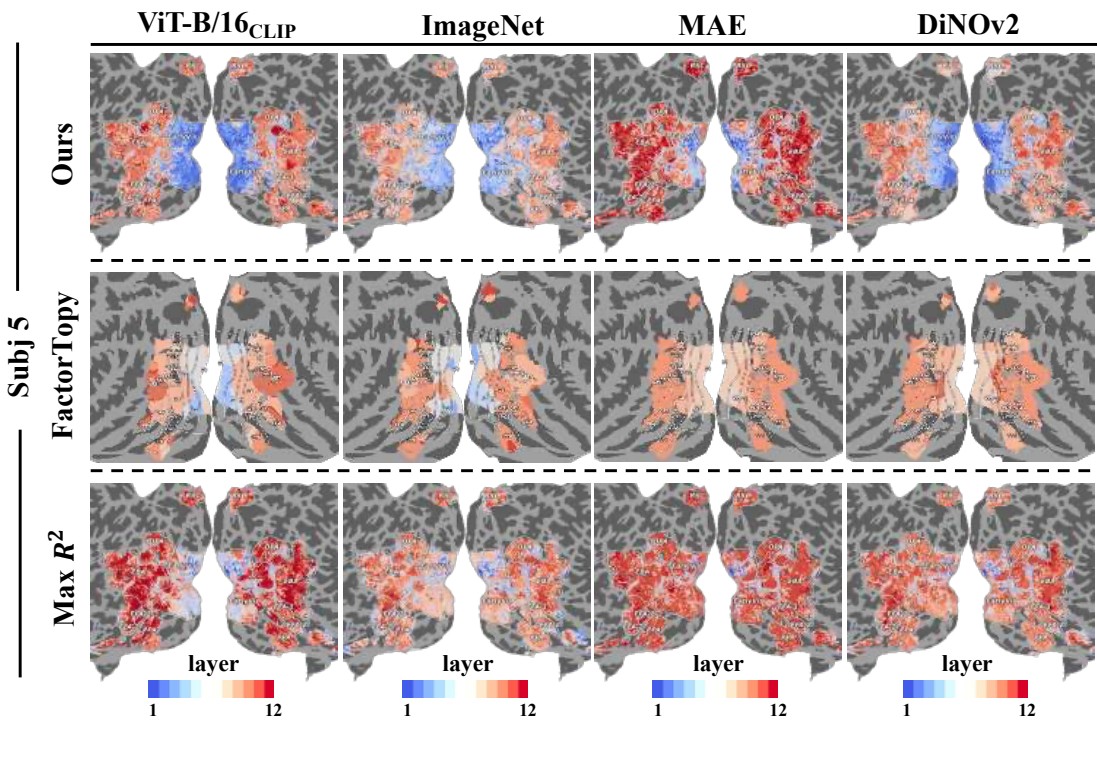

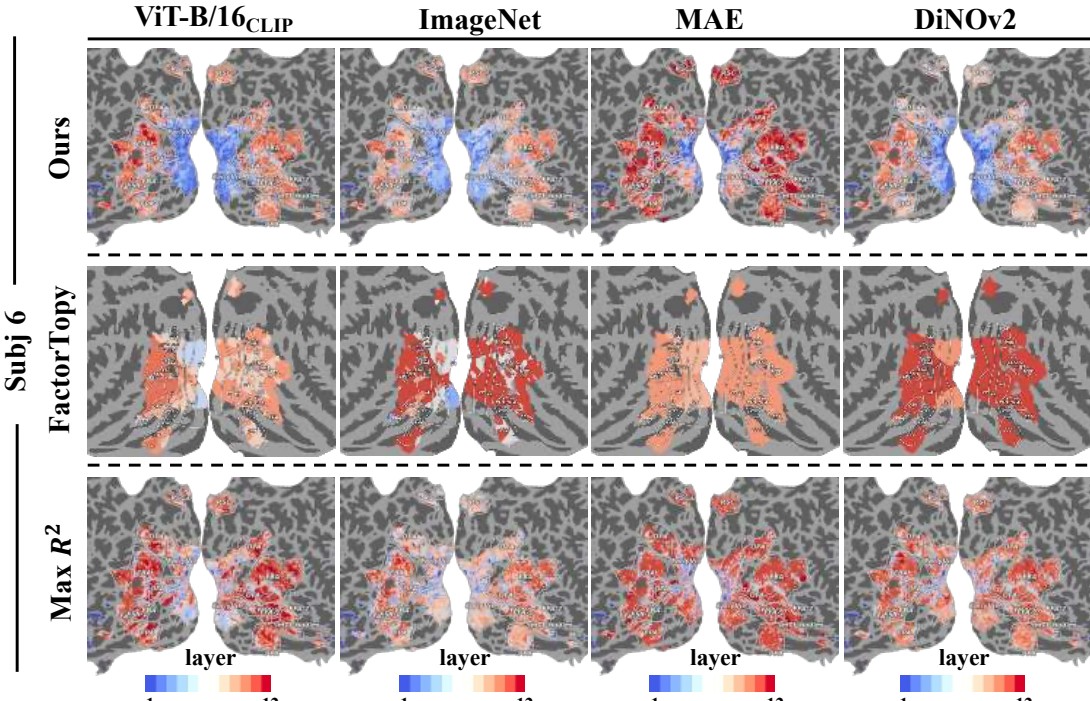

*Figure 17.* Top: Layer alignment for S5,S6 based on SAE-BrainMap, FactorTopy (Yang et al., 2024) and Max $R^2$ (Wang et al., 2023), we visualize five models with attention structure and have 12 layers.

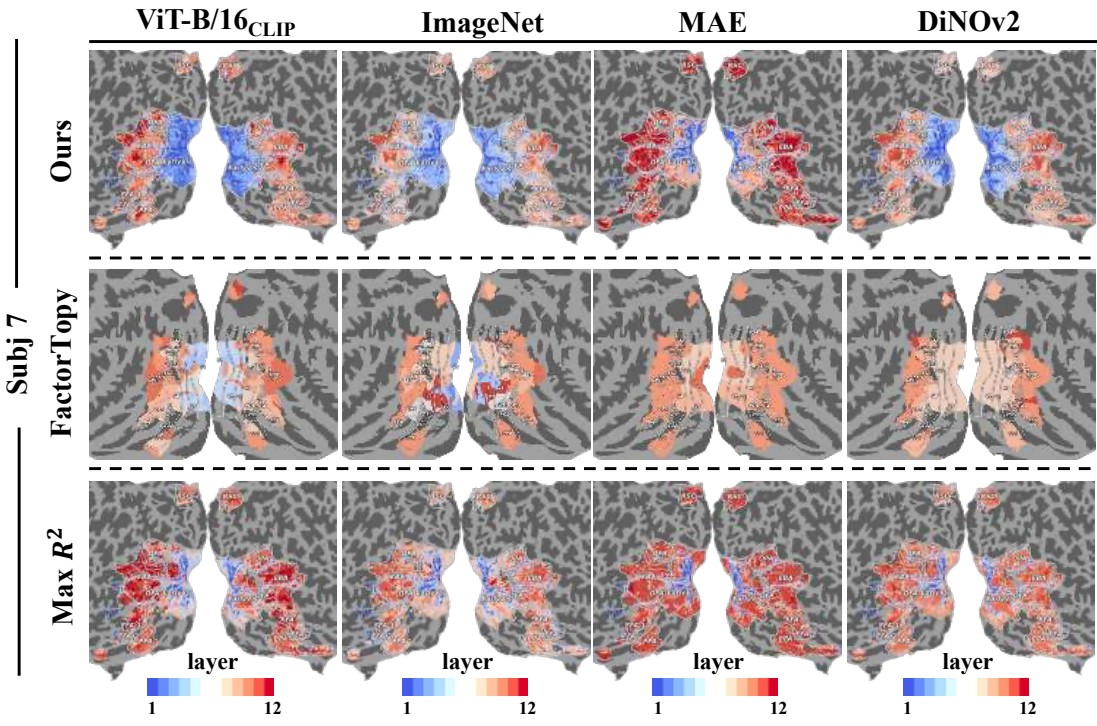

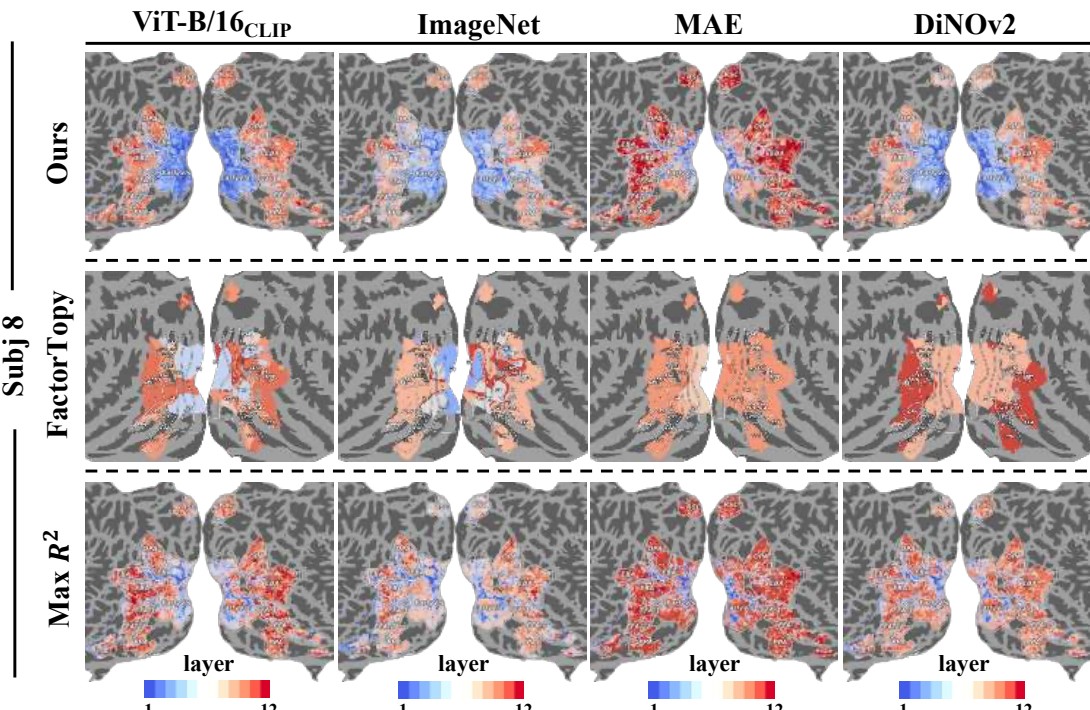

*Figure 18.* Top: Layer alignment for S7,S8 based on SAE-BrainMap, FactorTopy (Yang et al., 2024) and Max $R^2$ (Wang et al., 2023), we visualize five models with attention structure and have 12 layers.

