# OpenReview forum: "SAEs-BrainMap: Unveiling the Emergence of Specialized Concepts in Deep Models via Brain Alignment"
_ICML.cc/2026/Conference — ICML 2026 regular_

### Official Review · Reviewer_MGFs · 2026-03-03

**Soundness:** 3
**Presentation:** 3
**Significance:** 3
**Originality:** 3
**Overall Recommendation:** 4
**Confidence:** 4

**Summary:**

This paper proposes a novel framework, SAEs-BrainMap, which leverages human fMRI signals from the ventral visual pathway as objective probes to guide the identification and validation of features decomposed by SAE in deep vision models. The core idea is to invert the prevailing paradigm of "using models to explain the brain" into "using the brain to explain models." Through quantitative analyses, the authors demonstrate a robust representational alignment between SAE-derived features and activations in specific brain ROIs like FFA and RSC. They further utilize this alignment to trace the hierarchical emergence trajectory of generic visual concepts across model layers, offering novel, biologically-grounded insights into model interpretability.

**Compliance With Llm Reviewing Policy:**

Affirmed.

**Final Justification:**

ok

**Key Questions For Authors:**

Your current work aligns fMRI signals—either from regions of interest (ROIs) or individual voxels—with the activations of single neurons in large models, treating individual neurons as the minimal functional units. Have you considered that the true functional units in large models might be larger in scale, potentially consisting of coordinated ensembles of multiple neurons working together?

**Limitations:**

yes

**Strengths And Weaknesses:**

## Strength

- The proposed SAEs-BrainMap framework serves as a powerful methodological tool for future research aimed at systematically exploring the correspondence between artificial and biological neural systems.

- By mapping model layers onto the cortical surface, the work not only visualizes the model's global processing hierarchy but also reveals intriguing differences in concept learning strategies between architectures (e.g., CLIP vs. DINOv2), such as the earlier emergence of the 'face' concept in CLIP, which aligns with its image-text joint training objective.

## Weakness

- Equation (4) does not define what "Sim" refers to. Although I figured it out by examining the code, this is not clearly stated in the paper.

- The caption of Figure 10 in the appendix does not clearly explain what the figure illustrates.

- Methodologically, the paper is actually more aligned with the line of work on model-brain alignment, yet it lacks citations to key prior studies in this area. For example:

[1] "Coupling Visual Semantics of Artificial Neural Networks and Human Brain Function via Synchronized Activations," IEEE Transactions on Cognitive and Developmental Systems, 2023.

[2] "Coupling Artificial Neurons in BERT and Biological Neurons in the Human Brain," Proceedings of the AAAI Conference on Artificial Intelligence, 2023.

---

> ### Author Rebuttal · Authors · 2026-03-30
>
> We sincerely thank the reviewer for the positive feedback and thoughtful suggestions.
>
> **Q1: The meaning of “Sim” in Equation (4)**
> We thank the reviewer for pointing out this lack of clarity. In the revised version, we have explicitly defined how Sim is computed in the main text to avoid ambiguity.
>
> **Q2: The caption of Figure 10 in the appendix**
>
> We appreciate the reviewer’s comment. This figure presents the activation correlation between SAEs and fMRI signals for all 8 subjects as Fig.3A. The left cortex map denotes the maximum cosine similarity per voxel found across all models and layers, with additional plots showing the deviation of individual models relative to this peak. We will revise the caption to explain the figure more clearly in the revised version.
>
> **Q3: Prior studies**
>
> We appreciate the reviewer for bringing these relevant studies to our attention. We have carefully read the suggested papers and agree that they are closely related to our work. In the revised version, we have cited and discussed these studies to better position our method within the broader literature on model–brain alignment.
>
> **Q4: The combination of the functional units**
>
> We agree with the reviewer that semantic concepts are likely not represented by isolated features alone, but may instead emerge from the coordinated activity of multiple functional units. In this work, we start from single features as a tractable first step, because SAE features provide a natural sparse decomposition that makes feature-level analysis feasible and interpretable. While this does not fully capture distributed functional circuits, it offers a useful point for studying model function. We believe this is an important direction for future work, and we will further explore circuit-level mechanisms formed by multiple interacting features in subsequent research.
>
> We thank the reviewer again for the thoughtful and constructive feedback.

---

> > ### Author Rebuttal · Reviewer_MGFs · 2026-04-02
> >
> > OK

---

> > > ### Author Response · Authors · 2026-04-02
> > >
> > > We sincerely thank the reviewer for their constructive feedback. In the revised manuscript, we will implement the following changes:
> > >
> > > - We will incorporate the comparative experiments along with their detailed experimental settings into the revised manuscript.
> > >
> > > - We will provide a more comprehensive discussion of our work’s limitations in the Appendix, while more explicitly clarifying our core contributions in the main text.
> > >
> > > - We will carefully proofread the paper and correct all typographical and spelling errors.
> > >
> > > Once again, we greatly appreciate the reviewer’s positive and encouraging comments.

---

### Official Review · Reviewer_j8vY · 2026-03-03

**Soundness:** 3
**Presentation:** 3
**Significance:** 3
**Originality:** 3
**Overall Recommendation:** 5
**Confidence:** 3

**Summary:**

In this paper, authors use representations from the human visual stream to identify SAE features sensitive to semantic concepts such as face, food, words etc. The authors first show that SAE features are overall better aligned to brain activity than raw artificial model representations. Further, the authors show that the SAE features that are the most correlated to a specific ROI activations, also show high sensitivity to the specific concept associated with the ROI. Lastly, the authors use the number of associated semantic features per layer to show the emergence of the concept across the model layers. Faces have been shown to emerge earlier as compared to other concepts such as food or words.

**Compliance With Llm Reviewing Policy:**

Affirmed.

**Ethical Review Concerns:**

The authors have justified my questions with thorough ablations.

**Final Justification:**

The authors have justified my concerns with added ablation experiments.

**Key Questions For Authors:**

- Why have the authors not used linear predictivity as a brain alignment metric?
- The y axis label in figure 4C does not make sense to me and is confusing

**Limitations:**

My main concern/thought/question after reading this paper is whether the proposed approach offers a meaningful advantage over a simpler, established alternative: probing individual neurons directly in the model's raw activation space across layers.


While the authors demonstrate that SAE feature activations align more closely with brain data than raw model activations in aggregate, this comparison may be insufficient. The more informative baseline would be a direct comparison between (1) probing individual SAE features and (2) probing individual neurons in the model's native representational space. Without this comparison, it is unclear whether the insights we get (such as emergence of various concepts across model layers) stems from the SAE's sparse decomposition specifically, or simply from the act of identifying selective, interpretable units, which raw model neurons can also provide.

The authors motivate the use of SAEs by appealing to polysemanticity: because individual model neurons respond to multiple unrelated concepts, they argue that raw activations are not suitable for localization of fine grained concepts. However,even a polysemantic neuron that responds to both faces and food retains a selective response profile, and a probing analysis over such neurons could, in principle, recover the same information about when and where concepts are represented in the model. If polysemantic neurons still carry sufficient signal for this purpose, the added complexity of training and interpreting an SAE may not be justified.


The paper would be significantly strengthened by an ablation or direct comparison demonstrating that SAE features provide information that individual model neuron probing cannot, rather than assuming this advantage on theoretical grounds.

This is just food for thought. Needs not be included in the rebuttal experiments. An interesting avenue for future work would be to systematically characterize how polysemanticity evolves across model layers — both in the raw activation space and within the SAE feature space. Specifically, one could identify neurons (or SAE features) that respond selectively to multiple semantic categories and track how the prevalence and degree of such polysemanticity changes from early to later layers. This would shed light on whether deeper layers of the model become progressively more or less polysemantic?


Minor Corrections -

- Typo in line 91 - ‘interal’

- The caption for figure 4A and 4B seems to be interchanged

- I don't think Table 3 adds a lot to the paper

**Strengths And Weaknesses:**

Strengths - This paper presents a biologically grounded approach for identifying concept-specific semantic features in trained sparse autoencoders using brain activation data, and supports the proposed method with extensive ablation studies demonstrating its validity.

Weakness - Refer to the limitations

---

> ### Author Rebuttal · Authors · 2026-03-30
>
> We sincerely thank the reviewer for the thoughtful and constructive feedback.
>
> **Q1: Why have the authors not used linear predictivity as a brain alignment metric?**
>
> Our primary objective in establishing brain-model alignment is to demonstrate that empirical brain activations can serve as objective probes for selecting relevant model features, rather than training higher-performance encoding models, so it is neither necessary nor aligned with our goals. Furthermore, our preliminary attempts revealed that SAE activations are not suitable for directly training a linear encoding layer. This is due to the highly sparse nature of SAE representations, the limited available fMRI data (only 10,000 images per subject) and the massive parameter space required (approximately 100 million parameters for a single fully connected layer). Training such a predictive layer becomes computationally intractable
>
> **Q2: The y axis of Fig.4C**
>
> Figure 4C presents the results of the ablation study on the SAEs' reconstruction of input vectors. During the reconstruction process, we set the relevant features selected via brain activations to zero. If a feature is strongly correlated with the concept of the current input image, its removal causes a significant drop in the SAE's reconstruction performance. This performance degradation is measured by the Normalized MSE Value, which is plotted on the y-axis of Figure 4C. We have provided a detailed explanation in the latest manuscript.
>
> **Limitation**
>
> We agree with the reviewer's valuable suggestion. Accordingly, **we have compared our method against two baselines**, as illustrated in Figure 1 at the following anonymous link: [https://anonymous.4open.science/r/ICML26_SAEs-BrainMap_Rebuttal-2F8C]. The CLIP-Dissect [1] method, visualizes neurons' selectivity via the same text prompts mentioned in Appendix, whereas CLIP-SAEs method selectes SAEs features through an image-text comparison matrix as CLIP-Dissect. The experimental results demonstrate that our method is capable of extracting face-related selective features earlier, more accurately, and with significantly greater monosemanticity.
>
> Regarding the polysemantic nature of neurons, the reviewer points out that with precise neural dissection, we could similarly locate where concepts, like faces and food, are represented in the model. We agree with this assessment. However, doing so would likely require shifting away from top-activation analysis toward a more comprehensive evaluation of individual neuron selectivity.
>
> This is because the reviewer overlooks a crucial factor: high-level visual semantic information inherently encompasses numerous low-level semantic concepts. As demonstrated in our supplementary experiments, the highest activation selectivity of neurons in the model's early layers is primarily associated with low-level features such as textures and colors. To identify high-level semantic information within these early-layer neurons, it would be strictly necessary to record and analyze a much broader range of their activation values. Furthermore, even if images containing high-level semantics are found under non-top activation conditions, it becomes exceedingly difficult to prove whether the neuron is genuinely activated by the high-level semantic concept itself, or merely by the low-level features naturally embedded within it. In the visual domain, these features are intrinsically fused together within an image and cannot be neatly decoupled into discrete units like tokens in natural language.
> Regarding the evolution of polysemanticity, if we continue to rely on the top-activated method to determine neuron selectivity, our supplementary results indicate that neurons may exhibit even stronger polysemantic behaviors. However, we acknowledge that this is currently a preliminary analysis, and we have not yet conducted a more comprehensive and detailed investigation into this phenomenon.
>
> Minor Corrections
> - Typos and Captions: We will correct the spelling errors throughout the manuscript. We also sincerely thank the reviewer for catching the caption error in Figure 4.
>
> - Table 3 Placement: Our original intent for including Table 3 in the main text was to demonstrate the stability of our experimental results across different subjects. Based on your feedback, we will move Table 3 to the Appendix and replace it in the main text with the baseline comparison from our supplementary experiments.
>
> We thank the reviewer again for the thoughtful and constructive feedback.
>
> [1] Oikarinen, T. and Weng, T.-W. CLIP-Dissect: Automatic Description of Neuron Representations in Deep Vision Networks, June 2023.

---

> > ### Author Rebuttal · Reviewer_j8vY · 2026-04-01
> >
> > Thank You. The ablation experiment was quite helpful.

---

> > > ### Author Response · Authors · 2026-04-02
> > >
> > > We sincerely thank the reviewer for their constructive feedback. In the revised manuscript, we will implement the following changes:
> > >
> > > - We will incorporate the comparative experiments along with their detailed experimental settings into the revised manuscript.
> > >
> > > - We will provide a more comprehensive discussion of our work’s limitations in the Appendix, while more explicitly clarifying our core contributions in the main text.
> > >
> > > - We will carefully proofread the paper and correct all typographical and spelling errors.
> > >
> > > Once again, we greatly appreciate the reviewer’s positive and encouraging comments.

---

### Official Review · Reviewer_gqgC · 2026-03-09

**Soundness:** 3
**Presentation:** 3
**Significance:** 3
**Originality:** 4
**Overall Recommendation:** 5
**Confidence:** 4

**Summary:**

The paper proposes a framework for interpreting visual deep neural networks by aligning their internal representations with human brain activity. The approach trains layer-wise Sparse Autoencoders (SAEs) on vision models to extract sparse monosemantic features, and then uses fMRI activations from regions in the ventral visual pathway as probes to identify features that correspond to specific semantic concepts (e.g., faces, bodies, scenes). By analyzing the alignment between SAE features and brain regions, the method traces how high-level visual concepts emerge across layers of deep models and maps model layers to cortical regions.

**Compliance With Llm Reviewing Policy:**

Affirmed.

**Final Justification:**

I thank the authors for their further clarifications and responses. Overall, although I believe that some limitations still remain (which the authors agreed to explicitly refer to in the final version), the paper tackles a very interesting task, proposes a novel way to address it, and provides multiple analyses and validations. Therefore, while acknowledging the limitations of this work, I do believe it makes a significant contribution to the field, and I therefore change my recommendation to accept.

**Key Questions For Authors:**

The paper is interesting and may have potentially valuable contributions. However, the authors should clarify the exact contributions of the work and provide stronger validation, as suggested above. I would be willing to increase my score if these points are addressed, in particular by clearly explaining the main contributions and by adding proper validation against at least some of the suggested baselines.

**Limitations:**

yes

**Strengths And Weaknesses:**

## Strengths
* The paper tackles an interesting task in interpretability of neural networks.
* The idea of using brain region functionality to identify interesting and relevant SAE concepts is interesting.
* Visualizations are well done and show interesting results.
* Many interesting analyses are presented.

------

## Weaknesses

**1. Contribution is not well defined.**
It remains unclear what concrete insights are obtained for model interpretability. The authors should better position their work within the existing literature and clarify what their method contributes relative to prior work. While the idea of using brain signals is interesting, it is not clear in what way this approach is beneficial, and more specifically how it provides advantages compared to other possible approaches.

**2. Contribution is not well validated.**
The only baseline provided is the comparison with raw neuron activations. However, there is no comparison with alternative methods for identifying interpretable features. For example, several simple baselines could be considered:
a. Clustering SAE features and identifying concepts based on cluster structure.
b. Measuring correlation between SAE features and CLIP text embeddings of concepts (or the average CLIP/DINO embeddings of images belonging to those concepts), instead of using brain signals.
c. Applying the scoring procedure used in the paper to the top activating images of each SAE feature and evaluating concept alignment directly.

These simple baselines would help determine whether the use of brain data actually contributes additional value. Moreover, comparisons with previous work studying model representations should be included, or at least an explanation provided for why such comparisons are not made.

**3. Contribution to brain–machine learning interpretability should be clarified.**
Although the authors do not explicitly frame this as their main contribution, several analyses appear to reveal more about brain representations and the alignment between brain regions and deep model layers than about the models themselves. This could be an interesting contribution, but the paper does not clearly position it relative to existing work in this direction. As a result, it is difficult to evaluate the novelty or impact of this aspect. The authors should better situate their work within prior research and include additional experiments or analysis to clarify this contribution.

**4. Related work should be improved.**
a. The paper presents itself as a method for identifying relevant SAE features, yet the introduction and related work provide very little discussion of existing approaches for this task.
b. Two additional papers relevant for interpretability and inspiration should be considered and cited:
   * [1] Multidimensional feature tuning in category-selective areas of human visual cortex (Van Dyck et al., 2025).
   * [2] BrainExplore: Large-Scale Discovery of Interpretable Visual Representations in the Human Brain (Wasserman et al., 2025), which trains SAEs to discover visual representations in brain signals.

**5. Clarity and organization issues.**
Some parts of the paper should be written more clearly. A figure that explicitly presents the pipeline of the proposed method would improve clarity. While Figure 1 provides a high-level overview, a more detailed diagram of the method would help readers understand the pipeline. The results section is also difficult to follow. The text describing the results is often vague, making it hard to understand the key takeaways from each subsection and how they relate to the paper’s claims. Although the figures are visually appealing and potentially interesting, there are many of them, and in several cases it is unclear how they contribute to the central story of the paper. The authors should reorganize the results section to clearly state the main claims and show how each experiment supports them.

------

## Minor Comments
1. In the SAE architecture used in this paper, the encoder and decoder share the same linear weights. Please clarify why this design choice was made. In many SAE interpretability works this is not the case.
2. The paper claims that SAEs outperform raw model neurons in terms of activation similarity across models. Is it originated because SAE has larger number of features/units (randomly having more features might give higher scores regardless of the content of those features)?
3. Subject 5 is used for many of the visualizations. Please explain why this subject was chosen or provide results for additional subjects (e.g., Subject 1 or multiple subjects).
4. Figure 7: The numbers appear outside the frame. It is also unclear why the feature IDs are necessary, as they do not seem to add meaningful information in the main figure (they could instead be reported in the appendix).

---

> ### Author Rebuttal · Authors · 2026-03-30
>
> We sincerely thank the reviewer for the thoughtful and constructive feedback. To address the concerns raised, we would like to explicitly clarify our primary contributions:
>
> - **Direct Alignment**: We are the first to demonstrate a fine-grained, direct correlation between brain activity and DNN representations, both activationally and functionally.
>
> - **Biological Probing**: We pioneer the use of brain activation signals as objective, data-driven probes for deep model functional analysis.
>
> - **Concept Emergence**: We prior provide a novel visual framework that traces the hierarchical emergence of generic visual concepts within DNNs.
>
> **Q1: Contribution is not well defined.**
>
> Our work introduces a novel interpretability framework designed to answer a fundamental question: how do deep learning models learn to understand generic concepts? This remains a significant challenge for existing methods such as VCC[1] and GCC[2]. Those approaches primarily explain how models learn a specific, predefined class, heavily relying on target classification heads or specific image inputs as probes.
>
> Furthermore, regarding recent interpretability studies utilizing SAEs, current methods often rely on the model's final classification head weights as a prior to establish the connection between model functions and feature vectors. This weight-based approach **is fundamentally limited** when analyzing **intermediate layers**, which lack priors for their feature representations. By utilizing a purely activation-based method grounded in fMRI signals, our approach allows for unbiased analysis at any depth of the network.
>
> **Q2: Contribution is not well validate.**
>
> Thanks for the suggestion. To validate our approach, we implemented two baselines by utilizing a text-image similarity matrix as a probe to select both SAE features and raw neurons. We have visualized the comparative results in Fig.1 at the following anonymous link: [https://anonymous.4open.science/r/ICML26_SAEs-BrainMap_Rebuttal-2F8C]. This experiment underscores the distinct advantage of using biological signals to filter and isolate SAE features.
>
> **Q3: Contribution to brain–machine learning interpretability should be clarified.**
>
> Thanks for the suggestion. Given the novelty of this paradigm, we considered it important to first demonstrate whether biological signals can meaningfully characterize model functionality. To do so, we evaluated the method from three complementary perspectives: activational correspondence, structural correspondence, and functional consistency. As shown in paper's Fig3,4. These findings suggest that brain activity can serve as a useful and objective probe for model interpretability. We have provided a more detailed explanation in the revise version.
>
> **Q4: Related work should be improved.**
>
> We carefully reviewed both papers and appreciate the reviewer’s suggestion. We agree that these works are relevant to our study and will include them in the Related Work section in the revised version.
>
> **Q5: Clarity and organization issues.**
>
> The paper is organized around a two-stage logic. We first demonstrate that brain activations can serve as effective probes for investigating model functionality, and then present the analyses and findings made possible by this framework. In the revised version, we will update Fig.1 and reorganize the paper accordingly.
>
> **Minor Comments:**
>
> 1: **SAEs Structure**. Before we start our experiments, we trained both topk-SAEs and Vallina SAEs, the topk-SAEs highest average similarity is slightly lower than Vallina SAEs, with highest average similarity of 0.2693 and 0.2708. So we use Vallina SAEs in the experiments.
>
> 2: **SAEs' performance**. We believe that the higher correlation primarily stems from the greater monosemantic selectivity of SAE features. As illustrated in Fig.2 in anonymous link, raw neurons can be activated by semantically unrelated stimuli. We agree that increasing the number of features may also contribute to improved correlation to some extent; however, we do think this is the primary reason for the observed gain.
>
> 3: **The use of S5**. We mainly used Subject 5 for visualization, as in the other papers[3], while the results for the other subjects were included in the supplementary materials.
>
> 4: **Feature Number**. In the revised version, we will improve the corresponding figures and move the detailed feature ID records to the appendix.
>
> We thank the reviewer again for the thoughtful and constructive feedback.
>
> [1] Kowal, M. et al. Visual Concept Connectome (VCC): Open World Concept Discovery and their Interlayer Connections in Deep Models, April 2024.
> [2] Kwon, D. et al. Granular Concept Circuits: Toward a Fine-Grained Circuit Discovery for Concept Representations, August 2025.
> [3] Wang, A. Y. et al. Better models of human high-level visual cortex emerge from natural language supervision with a large and diverse dataset. Nature Machine Intelligence, November 2023.

---

> > ### Author Rebuttal · Reviewer_gqgC · 2026-04-01
> >
> > ## Updated Acknowledgment
> > I thank the authors for their further clarifications and responses. Overall, although I believe that some limitations still remain (which the authors agreed to explicitly refer to in the final version), the paper tackles a very interesting task, proposes a novel way to address it, and provides multiple analyses and validations. Therefore, while acknowledging the limitations of this work, I do believe it makes a significant contribution to the field, and I therefore ***change my recommendation to accept***.
> >
> > ---
> > ## Previous Acknowledgment
> >
> > I thank the authors for their response to my concerns. The authors addressed some of my concerns, and I am willing to change my score to accept after clarification of the following points:
> >
> > 1. **Tone down over-claiming regarding novelty.**
> > The claim of being the first to demonstrate a fine-grained, direct correlation between brain activity and DNN representations is not accurate. Correlations between brain activity and DNN representations have been studied for many years, with extensive prior work showing similarities in processing and feature representations between CNNs and the brain. This includes efforts such as the Brain-Score platform and works by researchers such as Martin Schrimpf, Galit Yovel, James J. DiCarlo, and others. The authors should better position their contribution within this existing line of work rather than claiming to be the first.
> >
> > 2. **Acknowledge the limitation of restricted concepts used.**
> > The authors should acknowledge as a major limitation that their method is applied only to a small set of predefined concepts (corresponding to well-known brain regions). This should be clearly stated and discussed in relation to prior work. The claim that previous methods only learn specific predefined classes is not sufficiently justified in the related work, and I am not convinced it is accurate. There are multiple works that analyze intermediate representations in LLMs, vision models, and VLMs (e.g., work by Yossi Gandelsman and others). Moreover, the concepts used in this paper are relatively broad and limited, which makes the overall contribution less clear.
> >
> > 3. **Clarify the new results and selection procedure.**
> > A clearer explanation is needed regarding the new results and how the best units were selected in each baseline. This is not clear from the rebuttal or from the attached material. In particular, it is unclear whether CLIP similarity is computed over the top images of each SAE feature neuron. A better understanding of this procedure is necessary to properly evaluate the contribution.

---

> > > ### Author Response · Authors · 2026-04-02
> > >
> > > We sincerely appreciate the Reviewer's prompt and thoughtful follow-up. We will carefully take these comments into consideration in the revision.
> > >
> > > **Q1: On over-claiming regarding novelty.**
> > >
> > > We fully agree with the Reviewer that our original wording inadvertently overstated the novelty of our work. We acknowledge that correspondence studies between brain activity and model activations are not new, and prior research—particularly those utilizing CNN architectures—has established a critical foundation in this domain.
> > >
> > > To clarify our specific contribution: while the granularity of aligning brain voxels has been explored before, our work is the first to leverage SAEs features to establish this fine-grained alignment (model features and brain voxels) within modern ViT architectures in this scale. Our experiments demonstrate that fine-grained feature-level representations can maintain activation, structural, and functional alignment with the brain.
> > >
> > > We thank the Reviewer for this important reminder, and we will revise the relevant text in the new version accordingly.
> > >
> > > **Q2: Acknowledge the limitation of restricted concepts used.**
> > >
> > > We agree with the Reviewer regarding this limitation. In particular, the lack of fine-grained functional priors at the voxel level currently constrains our analysis to five brain regions with relatively well-established functional selectivity. As a result, the scope of our study is limited to these predefined concepts.
> > >
> > > Moreover, we have corrected our previous imprecise phrasing regarding prior concept-based interpretability methods to eliminate potential misunderstandings. Our intended point was not that previous methods are limited to learning only predefined classes; rather, we have clarified that consistently localizing and tracking generic concepts across layers remains a significant challenge, particularly when aiming for both semantic coherence and cross-layer interpretability.
> > >
> > > **Q3: The details of the supplymental experiments.**
> > >
> > > We apologize for not describing this procedure clearly in our previous response.
> > >
> > > In the supplementary experiments, we added two comparison baselines. The first is CLIP-Dissect [1,2,3], which has been shown to be an effective method for interpreting neuron-level functions in vision models. The second is CLIP-SAEs, which follows a procedure similar to CLIP-Dissect, but is applied to SAE features instead of raw neurons.
> > >
> > > More specifically, we first compute the correlation matrix between image embeddings and text embeddings, denoted as $M \in \mathbb{R}^{[n, m]}$, where $n$ is the number of test images and $m$ is the number of text prompts provided in Appendix C. We then record the activations of all neurons or SAE features on the same image set (The ImageNet-1K test set as our work), denoted as $S \in \mathbb{R}^{[n, k]}$, where $k$ is the number of neurons or SAE features. Following CLIP-Dissect, we then compute the activation correlation between the image-text similarity matrix $M$ and the feature activation matrix $S$ as [1]. This produces a correlation matrix $C \in \mathbb{R}^{m, k}$, which measures the correlation between each feature and each text prompt. Finally, we select the features that are most strongly correlated with the prompt corresponding to the “face” concept, and visualize the feature with the highest correlation score. The displayed images are the top-4 that most activate the selected feature.
> > >
> > > We again sincerely thank the Reviewer for the thoughtful and constructive feedback.
> > >
> > > [1] Oikarinen, T. et al. CLIP-Dissect: Automatic Description of Neuron Representations in Deep Vision Networks, June 2023.
> > >
> > > [2] Bai, N. et al. Describe-and-Dissect: Interpreting Neurons in Vision Networks with Language Models, March 2024.
> > >
> > > [3] Yang, G. et al. CLIP-MSM: A MultiSemantic Mapping Brain Representation for Human High-Level Visual Cortex. Proceedings of the AAAI Conference on Artificial Intelligence, 39(9):9184–9192, April 2025. ISSN 2374-3468. doi: 10.1609/aaai.v39i9.32994

---

### Official Review · Reviewer_Xh3e · 2026-03-14

**Soundness:** 3
**Presentation:** 2
**Significance:** 2
**Originality:** 3
**Overall Recommendation:** 3
**Confidence:** 3

**Summary:**

Images in a constructed SAE space, and representations from the visual cortex are compared. Extensive analysis effort is made to align both representations. They find better results usinf SAE features than representations directly.

**Compliance With Llm Reviewing Policy:**

Affirmed.

**Final Justification:**

I believe their is a question of methodology and novelty which is complicated to justify. As thorough as this work is, the chosen approach and empirical results are made of hard to validate components. It it appears to me that the explanations needed to clarify this require a significant update to the paper.

**Key Questions For Authors:**

(1) It appears SAE space correlates better than pure latent space activation with brain voxels. By selecting specific features, the authors remove the noise of non-correlated features in the AI space. Do you have an intuition of what the added correlation from SAEs tells us about the geometric space of vision models? or perhaps in general what do SAEs bring that representations did not?

(2) How stable are your SAE concepts? I recently stumbled on the following work, Bolukbasi et al. (2021), "An Interpretability Illusion for BERT", basically sometimes SAE seem to be finding comcepts, but checking them out of domain reveals them to not be at all semantically significant. This is an issue for ealry layers of vision models particularly.

(3) Could you expend on what we learn from the different models having different correlations with brain regions?

**Limitations:**

yes

**Strengths And Weaknesses:**

Soundness: Extensive work, with complex concepts. There seems to have been extensive work and methodological effort at every step.

Soundness: Framing this work as interpretability has one drawback: we are improving understanding of a blackbox through another blackbox. On the positive side, this work takes a shortcut and avoids repeating all neuroscience experiments on vision models to get a similar mapping as the one we have of the vision ROIs in the brain. Nonetheless, Of the two, I would consider the brain to be the most opaque blackbox, the experiments that can be done on a brain can be done on a model, while the reverse is not true.

Presentation: There are some difficult language points, typos are minor, some sentences are difficult to process.

Significance:
I do not understand why we're looking through SAE space rather than directly at representation space.
My understanding is that the added value of SAEs is in how in divides the space into separate concepts. But if correlation with representation is already established, I do not understand the added value of concepts. Increasing correlation is not per se something we're seeking here. Here the conclusions are similar to previous works - process correlates, representations correlates - and the added value of SAEs seems under-used.

Originality: I have not heard of such work being done, especially at this scale.

---

> ### Author Rebuttal · Authors · 2026-03-30
>
> We sincerely thank the reviewer for the thoughtful and constructive feedback.
>
> **Q1: The use of Brain for Model interpretability**
> Although certain complex cognitive functions of the brain remain a 'black box,' the neural mechanisms underlying concept categorization in the higher-level ventral visual cortex have been supported by extensive evidence. We believe that while both artificial intelligence and the human brain operate as black boxes, mechanistic insights gained from either field can enhance our understanding of the other, thereby driving the mutual advancement of AI and neuroscience. In our work, we apply rigorous analysis and demonstrate that, at the ROI level, the brain can serve as a useful tool for probing models' function.
>
> **Q2: Why we look through SAE space**
>
> First, the correlation between models and the brain in our work is established through activation patterns, rather than through direct comparison in a shared vector representation space. Because we can not dirrectly find the vector representation for brain and the semantic concept for the vector in model's feature space. The key role of SAEs, therefore, is to provide vector representations for unknown monosemantic features in the model’s feature space, even though the exact meaning of these vectors may not always be fully interpretable to humans.
>
> Our goal is to further investigate model function. In our experiments, we do observe a certain degree of activation correlation between individual neuron activations and brain signals. However, **such correlation alone does not provide deeper insight into model function**, because **single neurons are often polysemantic**. As a result, activation-level correlation between model and brain is insufficient for more fine-grained functional analysis. This is precisely where SAEs are useful: as the reviewer pointed out, they divide the model’s representation space into more functionally separated concepts, and these concepts can often be visualized and interpreted.
>
> Our work connects vector representations with concepts through their activation patterns. We believe that the vector representation of the same concept may evolve across layers, while preserving a similar activation pattern. If we can establish a link between the activation of SAE-derived features and their semantic content, then we can, to some extent, build a bridge between vectors and semantics, which in turn allows us to probe the model’s feature space more meaningfully. In this process, the brain provides semantic–activation correlations. In our paper, we further support this idea through extensive experiments from the perspectives of activation, structure, and function, demonstrating the feasibility of using brain activity to probe model function.
>
> Therefore, the primary contribution of utilizing SAEs in our work extends beyond merely improving correlation; it fundamentally enables us to elucidate how concepts emerge across layers.
>
> **Q3: The advantage of SAEs**
>
> In the supplementary experiments, we compared our results with an alternative approach based on individual neurons; the results are shown in Supplementary Fig.1. The experiments indicate that, for functional analysis, SAEs can identify relevant features at earlier stages. Moreover, due to their more monosemantic nature, the resulting functional visualizations are easier for humans to interpret. In Supplementary Fig.2, we further visualize the polysemantic nature of model neurons. The anonymous link to our supplementary experiments is provided below: [https://anonymous.4open.science/r/ICML26_SAEs-BrainMap_Rebuttal-2F8C]
>
> **Q4: How stable are your SAE concepts**
>
> We recognize that features decomposed by SAEs can sometimes still be difficult for humans to interpret. To address this, after utilizing brain activations to filter the top 100 most relevant features, we incorporated rigorous stability and semantic alignment analyses. The line chart illustrating the count of stable features within this top-100 subset is presented in Figure 4A of the manuscript. Furthermore, following the extraction process, we manually inspected the selectivity heatmaps of these SAEs features and find they can be described using natural language.
>
> **Q5: What we learn from the experiments**
>
> Our experiments reveal a finding: given comparable parameter scales and similar architectures, training on larger datasets and incorporating supervisory signals tend to facilitate the earlier emergence of high-level semantic concepts in shallower layers. Furthermore, the correspondence between model layers and the high-level visual cortex highlights a functional divergence: while the final two layers of most models are typically specialized for their respective downstream tasks, the unique training paradigm of MAE results in its top-most layers maintaining a strong representational alignment with high-level biological visual areas.
>
> We thank the reviewer again for the thoughtful and constructive feedback.

---

> > ### Author Rebuttal · Reviewer_Xh3e · 2026-03-31
> >
> > Thank you for the clear answer, and again I would like to acknowledge the amount of work apparent in this manuscript.
> >
> > The authors rightfully remind me that:
> > (1) A blackbox CAN be a good way to look into another blackbox. (2) The reason SAEs are used is that it transforms a polysemantic space into a set of monosemantic features. (3) They clarify their findings on MAE, and on vision models
> >
> > The link to supplementary experiment does not exist/does not work. I will assume in good faith they exist and that their conclusions are valid but have no proof, and could not evaluate it.
> >
> > I believe I cannot raise the score I gave:
> >
> > > The blackbox is not inevitable, but has little advantage
> > Here, authors go through fMRI, then SAEs, then build a specific comparison protocol.
> > They do not repeat the experiments that led to the identification of ROIs in the brain - which is possible, and comes from neuroscience experiments already validated by the community. This would go a great way to validate the complex methodology they built empirically.
> > Theoretically fMRI + SAE + comparison approach does not appear to have an advantage to compensate for the uncertainty inherent to them. The blackbox usage has unsufficient advantage.
> >
> > > the findings summarized in Q5 seem to point at which model objectives most follow the brain's hierarchy.
> >
> > While the rebuttal clarifies arguments which were unclear to me from reading the paper, I am not convinced by the methodology's relevance and stability. I believe in the author's efforts to clarify this, but it appears to me that the explanations needed to clarify this require a significant update to the paper.
> >
> > I will nonetheless not decrease the score. Effort (should the anonymous link end up working) seems to have been made to test for some weaknesses such as noise of fMRI signal or instability of SAEs. while this fall short of validating the methodology, it goes in that direction.

---

> > > ### Author Response · Authors · 2026-04-01
> > >
> > > We sincerely thank the reviewer for their additional feedback and constructive insights. We would like to address the specific points raised as follows:
> > >
> > > **Correction of the Supplementary Link.**
> > >
> > > We apologize for the broken link in our previous submission. The issue was caused by a Markdown formatting oversight that inadvertently included a closing bracket ']' within the hyperlink syntax. The corrected, fully functional link to our supplementary materials is now available here: https://anonymous.4open.science/r/ICML26_SAEs-BrainMap_Rebuttal-2F8C
> > >
> > > **Blackbox vs fLoc Experiments**
> > >
> > > Again, we emphasize that, while the brain is of course not fully understood, the human ventral visual pathway is not an uncharacterized blackbox. Decades of cognitive neuroscience and systems neuroscience have revealed substantial and reproducible functional structure in this system, including hierarchical organization, category-selective regions, and stable representational principles that have been repeatedly validated across studies. Our work builds precisely on these accumulated findings. Furthermore, relevant research in the Neuroscience4AI community has demonstrated that this approach represents a viable and promising direction[1,2,3].
> > >
> > > Regarding the suggestion to apply fLoc experiments[4]—typically used to identify brain functional regions—directly to models to reveal their functional organization: We agree that this is a valuable and theoretically. However, conducting fLoc experiments on the model requires a significantly larger and more finely grained dataset, likely necessitating the extraction of semantic masks—a highly labor-intensive endeavor. Furthermore, recording model activations across diverse conditions while rigorously controlling for confounding variables represents a methodological challenge that took decades to refine in neuroscience. In contrast, a key advantage of the black-box approach lies in its ability to effectively leverage domain priors, enabling rapid and precise identification of model representations, thereby facilitating the scalable extension of experiments to larger cohorts.
> > >
> > > **Methodological Relevance and Stability**
> > >
> > > - Regarding fMRI Signal Quality: We acknowledge that the signal-to-noise ratio (SNR) of fMRI data is an inherent limitation of our approach. However, the dataset we utilized, NSD, has been extensively adopted in numerous prior studies for voxel-level functional analysis of the human brain [5,6,7]. This widespread usage demonstrates that the SNR of this dataset is sufficiently robust for high-resolution neuroimaging research. Furthermore, as explicitly stated in our paper, we deliberately chose to conduct our functional validation at the Region of Interest (ROI) level rather than the voxel level specifically to mitigate concerns regarding SNR and to ensure more reliable statistical power.
> > >
> > > - Regarding SAE Feature Stability: We recognize that not every feature extracted by Sparse Autoencoders (SAEs) possesses clear semantic interpretability. Nevertheless, our validation results demonstrate that the majority of the features we selected are stable and significantly enhance our understanding of the model's cognitive processing flow. At the current level of analysis, the performance of SAEs proves sufficient for our objectives.
> > >
> > > Once again, we appreciate the reviewer's thoughtful comments.
> > >
> > > [1] Xue, M. et al. A Convolutional Neural Network Interpretable Framework for Human Ventral Visual Pathway Representation. Proceedings of the AAAI Conference on Artificial Intelligence, 38(6):6413–6421, March 2024. ISSN 2374-3468. doi: 10.1609/aaai. v38i6.28461.
> > >
> > > [2] Yang, H. et al. Brain decodes deep nets. In Proceedings of the IEEE/CVF Conference on Computer Vision and Pattern Recognition (pp. 23030-23040).
> > >
> > > [3] Zhuang, C. et al. 2021. Unsupervised neural network models of the ventral visual stream. Proceedings of the National Academy of Sciences, 118(3): e2014196118.
> > >
> > > [4] Stigliani, A. et al. Temporal processing capacity in high-level visual cortex is domain specific. J. Neurosci. 35, 12412–12424 (2015).
> > >
> > > [5] Luo, Andrew F., et al. "Brainscuba: Fine-grained natural language captions of visual cortex selectivity." arXiv preprint arXiv:2310.04420 (2023).
> > >
> > > [6] Yang, G. et al. CLIP-MSM: A MultiSemantic Mapping Brain Representation for Human High-Level Visual Cortex. Proceedings of the AAAI Conference on Artificial Intelligence, 39(9):9184–9192, April 2025. ISSN 2374-3468. doi: 10.1609/aaai.v39i9.32994
> > >
> > > [7] Aria Y. Wang, et al. Better models of human high-level visual cortex emerge from natural language supervision with a large and diverse dataset. Nature Machine Intelligence, 5(12):1415–1426, November 2023. ISSN 2522-5839. doi: 10.1038/ s42256-023-00753-y.

---

### Decision · Program_Chairs · 2026-04-30

**Decision:**

Accept (regular)

**Comment:**

The paper studies interpretability of deep vision models by aligning sparse autoencoder (SAE) features to human fMRI responses. The conceptual twist is to use brain measurements as an external probe of model internals, instead of only using models to predict brain activity. The analysis traces how concept-like structure emerges across layers and how that emergence relates to visual hierarchy in cortex.

Reviewers found the direction intellectually fresh and aligned with current interest in mechanistic interpretability. Using fMRI as a grounding signal is methodologically nontrivial and the paper provides concrete analyses across models and layers. The empirical story benefits from comparing SAE features to raw activations, which was a key reviewer request.

Initial reviews asked whether SAEs are essential or whether raw activations already tell the same story, and requested stronger ablations and clearer statements of novelty relative to prior brain-model alignment work. The rebuttal added targeted comparisons showing SAE features can be more monosemantic or better aligned depending on the metric, which moved several reviewers to fully resolved. One reviewer remained partially unresolved on framing and contribution boundaries: they wanted sharper demarcation from prior alignment work and clearer limits on what the fMRI probe can and cannot prove about “concepts” in models.

Multiple reviewers marked fully resolved. One reviewer remained partially unresolved or unresolved on scope of claims, but the majority trend was positive.

Minor: careful language about causality and about what “brain alignment” means statistically (correlation vs mechanism).

Despite the one mixed acknowledgement, the paper is broadly seen as a strong interpretability contribution. Given these strengths, we accept.